# Perceptual Flow Network for Visually Grounded Reasoning

Yangfu Li [1 2 * #]   Yuning Gong [3 4 * #]   Hongjian Zhan [1 5 §]   Teng Li [2 6 #]   Yuanhuiyi Lyu [2 7 #]   Tianyi Chen [8]   Qi Liu [1]
Ziyuan Huang [2]   Zhihang Zhong [8 §]   Dandan Zheng [2 §]   Yue Lu [1 §]

## Abstract

Despite the success of LVLMs, general optimization objectives (*e.g.*, standard MLE) fail to constrain visual trajectories, leading to language bias and hallucination. To mitigate this, current methods introduce geometric priors from visual experts as additional supervision. However, we observe that such supervision is typically suboptimal: *it is biased toward geometric precision and offers limited reasoning utility*. To bridge this gap, we propose Perceptual Flow Network (PFlowNet), which eschews rigid alignment with the expert priors and achieves interpretable yet more effective visual reasoning. Specifically, PFlowNet decouples perception from reasoning to establish a self-conditioned generation process. Based on this, it integrates *multi-dimensional rewards* with *vicinal geometric shaping* via variational reinforcement learning, thereby facilitating reasoning-oriented perceptual behaviors while preserving visual reliability. PFlowNet delivers a provable performance guarantee and competitive empirical results, particularly setting new SOTA records on V* Bench (90.6%) and MME-RealWorld-lite (67.0%).

## 1. Introduction

Large Vision-Language Models (LVLMs) extend pretrained Large Language Models (LLMs) by integrating sophisticated vision encoders (Radford et al., 2021) and cross-modal alignment (Liu et al., 2023), achieving remarkable performance across diverse visual tasks (Bai et al., 2025a;b; Liu et al., 2024a). However, LVLMs still face challenges with interpretability and hallucination, particularly in complex scenarios, *e.g.*, fine-grained visual understanding.

---
*Equal contribution. #Work done during authors' internships. [1]ECNU [2]Ant Group [3]SCU [4]Shanghai AI Laboratory [5]Chongqing Institute of ECNU [6]HKUST [7]HKUST (GZ) [8]SJTU. Correspondence to: Hongjian Zhan <hjzhan@cee.ecnu.edu.cn>, Zhihang Zhong <zhongzhihang95@gmail.com>, Dandan Zheng <yuandan.zdd@antgroup.com>, Yue Lu <ylu@cs.ecnu.edu.cn>.

*Proceedings of the 43rd International Conference on Machine Learning*, Seoul, South Korea. PMLR 306, 2026. Copyright 2026 by the author(s).

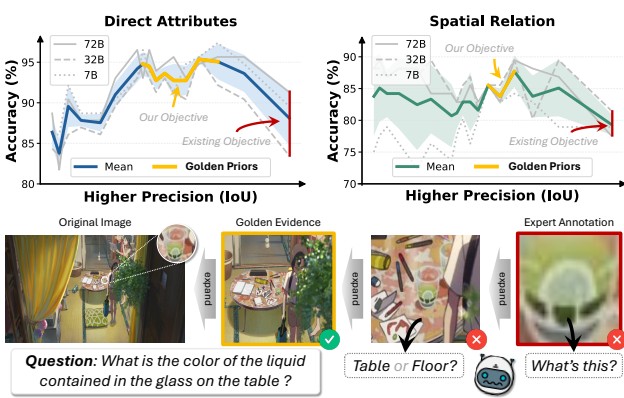

*Figure 1.* Impact of evidence geometric precision (IoU *w.r.t.* the ground truth) on reasoning performance (accuracy). The evidence with *minimum* and *maximum* precision is actually the full image and the expert annotation (*outlined in red*), respectively.

To enhance reliability, recent advances (Liu et al., 2025a; Wang et al., 2025a;b; Sarch et al., 2025; Liu et al., 2025b) distill geometric priors from visual experts, *e.g.*, GroundingDINO (Liu et al., 2024c), into LVLMs via *Reinforcement Learning with Verifiable Reward* (Guo et al., 2025b). By directly maximizing geometric consistency between LVLM predictions and expert priors, these approaches effectively anchor intermediate reasoning processes in visual evidence.

Despite this progress, a critical question remains:

> *The visual experts are initially designed for object detection; thus, are the geometric priors derived from these experts truly optimal for visual reasoning?*

**Preliminary Study.** To investigate this, we conduct a probing study using Qwen2.5-VL (Bai et al., 2025b) family on V* (Wu & Xie, 2024). This benchmark encompasses *direct attribute* recognition and *spatial relation* reasoning, backed by fine-grained expert annotations. We generate varying geometric priors by isotropically expanding the original annotations from their centers. By feeding the models directly with these evidence crops instead of full images, we measure the reasoning utility of different geometric priors. As illustrated in Figure 1, we observe a counterintuitive result: the most precise geometric prior, *i.e.*, expert annotation, is *not* the most helpful for reasoning. We attribute this to

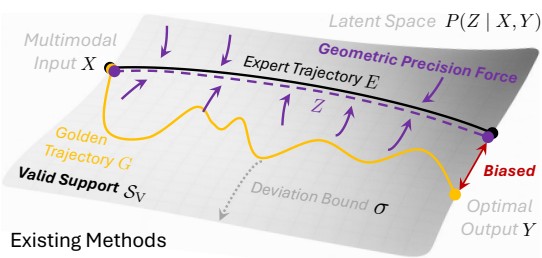 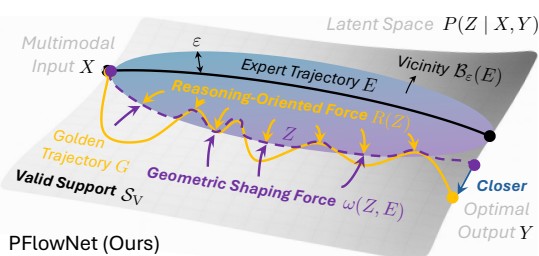

*Figure 2.* Illustration of feasible regions ($\mathcal{S}_\text{v}$) and optimization objectives for visually grounded reasoning. Existing methods constrain LVLMs to imitate expert trajectories by maximizing their geometric consistency, whereas PFlowNet integrates a reasoning-oriented reward with vicinal geometric shaping to achieve more sufficient yet controlled exploration, leading to reliable and high-efficacy reasoning.

a fundamental mismatch between the design principles of visual experts and LVLMs. While these experts are optimized to localize evidence with strict geometric precision, such an approach may induce a *tunnel vision* effect during reasoning. This effectively excludes context necessary for comprehensive understanding and degrades performance.

A natural intuition is to approximate the *golden* evidence by applying heuristic transformations to expert priors, thereby constructing less biased geometric guidance for LVLMs. However, we find that the optimal geometric prior is highly *instance-specific*, making such strategies intractable.

Motivated by this challenge, we propose Perceptual Flow Network (**PFlowNet**). Instead of constraining visual rationales via rigid alignment with static geometric priors, PFlowNet employs a *self-parameterized* variational distribution to approximate the posterior of idealized perceptual behaviors. By sampling from the optimized intrinsic distribution, PFlowNet *self-conditions* its subsequent reasoning process, yielding grounded yet more accurate outputs. To realize this, PFlowNet features three key innovations: ❶ **Perceptual Flow**, *i.e.*, a structured trajectory formulation, designed to effectively characterize perceptual behaviors in LVLMs, facilitating efficient optimization via hierarchical variational objectives, *e.g.*, Sub-Trajectory Balance (SubTB). ❷ **Tailored Framework** that efficiently achieves the *self-conditioned* autoregressive generation; and particularly, ❸ **Variational Reinforcement Fine-Tuning Strategy** that integrates a *multi-dimensional reward function* with a *vicinal geometric shaping* scheme to encourage reasoning-oriented perceptual behaviors while preserving visual reliability.

We provide theoretical analysis that establishes a provable performance guarantee for PFlowNet, as detailed in Theorems 3.1, 3.4. Moreover, comprehensive experimental results demonstrate its superiority across both general-purpose and fine-grained visual tasks from the empirical perspective. Importantly, it achieves substantial improvements of 13.1%, 10.4% and 21% over the base model (*i.e.*, Qwen3-VL 8B) on V* Bench, TreeBench, and MME-RealWorld-lite, respectively. Further analysis highlights its favorable performance-efficiency balance and effective test-time scaling properties.

## 2. Background and Motivation

### 2.1. Problem Formulation

Let $\mathcal{M}_\theta$ denote an LVLM parameterized by $\theta$, built upon a standard transformer architecture (Vaswani et al., 2017). Given a multimodal input $X$ (*e.g.*, images and instructions), $\mathcal{M}_\theta$ defines an autoregressive conditional distribution:

$$p_\theta(Y \mid X) = \prod_{t=1}^{T} p_\theta(y_t \mid X, y_{<t}),$$

where $Y = (y_1, y_2 \ldots, y_T)$ represents the output token sequence conditioned on $X$. Conventionally, $\mathcal{M}_\theta$ is optimized via Maximum-Likelihood Estimation (MLE):

$$\max_\theta \; \mathbb{E}_{(X,Y) \sim P_\text{data}}\big[\log p_\theta(Y \mid X)\big].$$

Despite the remarkable efficacy of this paradigm, it remains challenging to mitigate hallucination in $\mathcal{M}_\theta$ (Liu et al., 2024b; Gunjal et al., 2024; Chen et al., 2024), particularly in visual-centric applications (*e.g.*, fine-grained visual search). To formalize this, we consider the visual reasoning trajectory (*e.g.*, the sequence of RoIs) as a latent variable $Z$. In this view, the fundamental cause of hallucination stems from an ill-posed posterior $P(Z \mid X, Y)$ that may assign probability mass to *invalid* trajectories $Z$. Inspired by the success of RLVR in LLMs (Guo et al., 2025b), recent works explore incorporating geometric priors as verifiable rewards to constrain $Z$ for Visually Grounded Reasoning (VGR).

**Definition 2.1** (Visually Grounded Reasoning). Consider an input-output pair $(X, Y)$ and a *golden* visual trajectory $G$ that mediates the inference process $X \xrightarrow{G} Y$. We define $\mathcal{S}_\text{V}$ as the support of all *valid* visual trajectory $Z$, which is the $\sigma$-neighborhood of $G$ under a deviation metric $d(\cdot, \cdot)$:

$$\mathcal{S}_\text{V} \coloneqq \big\{ Z \mid d(Z, G) \leq \sigma \big\}.$$

*Target posterior* $P_V(Z \mid X, Y) \coloneqq P(Z \mid X, Y, Z \in \mathcal{S}_V)$ is given by assigning its probability mass exclusively to this support, and visually grounded reasoning is formulated as

$$\max_\theta \; \mathbb{E}_{(X,Y) \sim P_\text{data}}\big[\log \int_{\mathcal{S}_\text{V}} p_\theta(Y, Z \mid X) \, dZ\,\big]$$

which encourages $\mathcal{M}_\theta$ to not only yield the correct answer $Y$ but to anchor its latent visual rationales $Z$ to $G$.

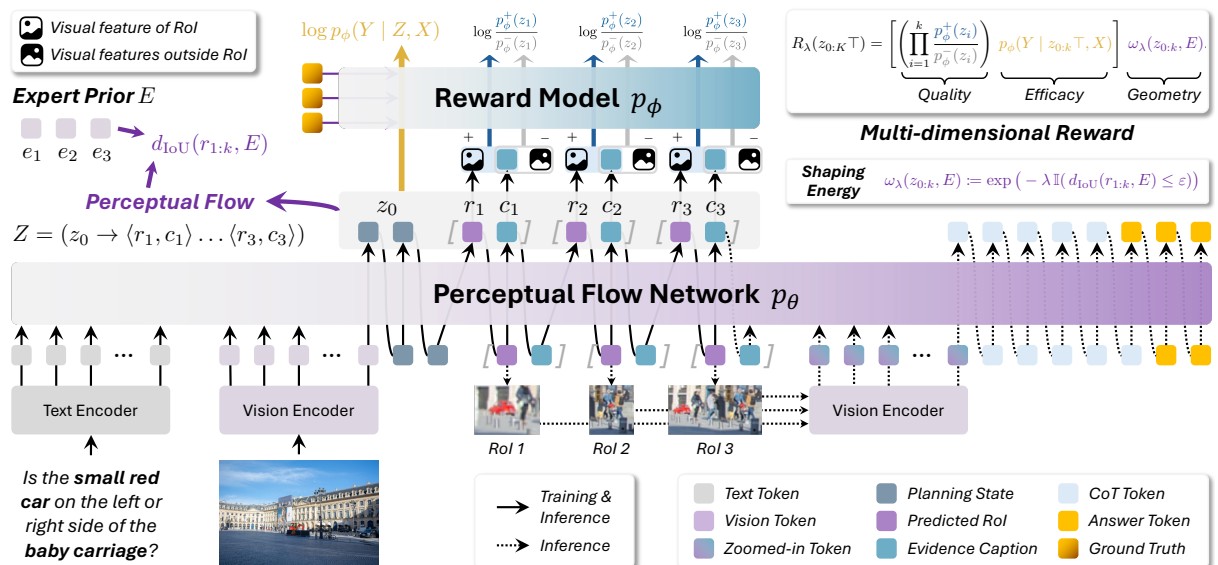

**Figure 3.** Overview of PFlowNet that consists of two *decoupled* stages: flow generation and flow-guided reasoning. We leverage a frozen reward model with the multi-dimensional reward to guide PFlowNet toward reasoning-oriented yet visually reliable perceptual flows. During reasoning, PFlowNet integrates the textual flow with corresponding visual features to derive interpretable and accurate answers.

## 2.2. Revisit VGR as Reasoning over Perceptual Flow

The golden trajectory $G$ is generally intractable; thus, previous works typically adopt well-trained visual experts, *e.g.*, GroundingDINO (Liu et al., 2024c) to synthesize a proxy for $G$. However, these experts are initially optimized for visual grounding rather than downstream visual reasoning. As a result, the synthetic trajectory is biased toward high geometric precision rather than reasoning utility, leading to suboptimal performance of the policy $p_\theta(Y|X)$, as revealed in Figure 2. To address this misalignment, we apply a *self-parameterized* variational distribution $p_\theta(Z \mid X)$ to approximate the target posterior $P_V(Z \mid X, Y)$, thereby achieving visually grounded reasoning via a latent-variable mixture:

$$p_\theta(Y, Z \mid X) = p_\theta(Z \mid X)\, p_\theta(Y \mid X, Z),$$

which is the key insight of the proposed PFlowNet. To more precisely characterize the behaviors of LVLMs and thus facilitate the optimization of $p_\theta(Z \mid X)$, we further introduce the concept of *Perceptual Flow* in the following.

**Definition 2.2** (Perceptual Flow). Given a multimodal input $X$, we define the **Perceptual Flow** $Z$ as a structured latent trajectory $Z = (z_0 \rightarrow z_1 \ldots z_K)$ that explicates the visual reasoning process. It comprises two distinct state types:

◇ **Planning State** ($z_0$): A language sequence enclosed by special tokens $\langle$analyze$\rangle$ and $\langle$/analyze$\rangle$. This state decomposes the query within $X$ and identifies relevant visual candidates for subsequent exploration.

◇ **Perceptual States** ($z_{\geq 1}$): A chain of grounded observations enclosed by $\langle$localize$\rangle$ and $\langle$/localize$\rangle$. Each state $z_k = \langle r_k, c_k \rangle$ consists of a Region of Interest (RoI) $r_k \in \mathbb{N}^4$ (represented in *absolute* coordinates) and a corresponding descriptive caption $c_k$.

Leveraging this design, we incorporate Sub-Trajectory Balance (Sub-TB) (Madan et al., 2023), a hierarchical variational objective in this work. Unlike standard RL paradigm, this formulation provides dense intermediate supervision, thereby facilitating diverse perceptual behaviors. Formally, given a perceptual flow $Z \sim p_\theta(Z \mid X)$, let $z_{i:j} \preceq Z$ be any sub-trajectory indexed by $0 \leq i \leq j \leq K$, the Sub-TB objective derived by a divergence metric $D$ is defined as:

$$\min_\theta \sum_{i,j} D\big(\mathcal{F}(z_i)\,\mathcal{T}_F(z_{i:j}) \,\|\, (\mathcal{F}(z_j)\,\mathcal{T}_B(z_{j:i})\big) \quad (1)$$

where $\mathcal{T}_F(z_{i:j}) = \prod_{k=i+1}^{j} p_\theta(z_k \mid z_{k-1})$ and $\mathcal{T}_B(z_{j:i}) = \prod_{k=i+1}^{j} p_\theta(z_{k-1} \mid z_k)$ denote the forward and backward transitions over the flow $Z$, respectively. $\mathcal{F}(z)$ is the total probability mass of all flows passing through the state $z$.

## 3. Perceptual Flow Network

The overall architecture of PFlowNet is shown in Figure 3. Formally, let $X \coloneqq \langle I, T \rangle$ denote a multimodal input consisting of an image $I$ and an instruction $T$. PFlowNet first samples the perceptual flow $Z$ from its *intrinsic* distribution and then yields the grounded output $Y$ via a *self-conditioned* generation. The joint distribution is factorized as:

$$p_\theta(Y, Z \mid X) = p_\theta(Z \mid X)\, p_\theta(Y \mid Z, \langle X, I_{\mathrm{RoI}} \rangle),$$

where $I_{\mathrm{RoI}}$ denotes the region of interest from the image $I$ conditioned on the perceptual flow $Z$. To effectively optimize the parameterized variational distribution $p_\theta(Z \mid X)$, we employ a progressive training paradigm. First, guided by the insights in Figure 1, we design a tailored data pipeline to synthesize fine-grained trajectories, explicitly aimed at preliminarily mitigating the inductive bias inherent in visual experts. Based on this, we bootstrap the model's capability

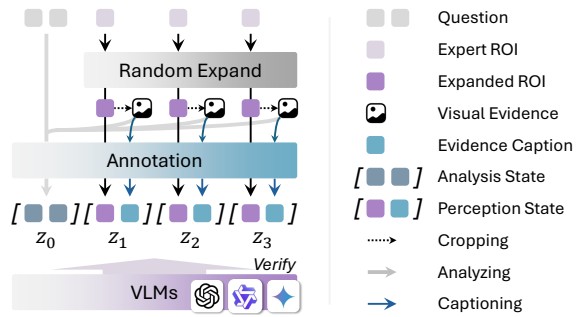

*Figure 4.* Illustration of the pipeline for perceptual flow synthesis.

to generate perceptual flows via Supervised Fine-Tuning (SFT). Furthermore, we propose a variational Reinforcement Fine-Tuning (RFT) strategy that integrates a carefully-designed reward and a vicinal geometric shaping to ensure a better approximation of the target posterior. This design liberates the model from the constraints of expert geometric priors, enabling it to extensively explore genuinely effective perceptual behaviors while maintaining visual reliability.

### 3.1. Training Data Curation and Cold Start

**Data Collection.** We curate high-quality training data for cold start and subsequent RFT based on three principles:

$\diamond$ **Diverse Tasks.** We consider a broad spectrum of visual tasks, spanning both fine-grained visual understanding and general-purpose scenarios, which ensures the model develops generalizable perceptual behaviors.

$\diamond$ **Diverse RoIs.** To prevent overfitting to specific spatial patterns, we perform cross-expert annotation for each sample and preserve the samples whose RoIs have sufficiently broad and diverse spatial coverage.

$\diamond$ **Appropriate Difficulty.** To maximize training efficiency, we exclude trivial samples that are readily solvable by the base model and target hard samples that significantly exceed the model's current capabilities to synthesize fine-grained supervision for SFT. Meanwhile, samples with moderate difficulty are reserved for the subsequent RFT.

**Flow Synthesis.** We construct cold-start datasets by eliciting step-by-step trajectories from teacher models, *e.g.*, Gemini3 (DeepMind, 2025a) and GPT-4o (OpenAI, 2024). As shown in Figure 4, for each sample with annotated RoI, we first randomly expand every RoI to mitigate the inductive bias introduced by visual experts. Each teacher is then prompted to (i) determine the critical visual content conditioned on both the question and RoI (serving as the *Planning State* $z_0$), and (ii) generate detailed captions for each piece of visual evidence. Each expanded RoI, together with its caption, is treated as a *Perceptual State* $z_{\geq 1}$. The perceptual flow is then formed by composing the planning state $z_0$ with all subsequent perceptual states $(z_1, z_2, \ldots, z_n)$.

Subsequently, we use verifier models to refine the synthetic flows $Z_s$ via rejection sampling and partition the retained

samples by difficulty into a cold-start dataset $P_{\text{SFT}}(X, Z_s)$ and an RL dataset $P_{\text{RFT}}(X, Y, E)$. Here, $X$ denotes the multimodal input, $Y$ is the *accepted response* of the verifier conditioned on $Z_s$, and $E$ denotes the expert-annotated RoIs. We then perform supervised fine-tuning on $P_{\text{SFT}}(X, Z_s)$ to equip the policy the ability to generate structured perceptual flows from the inputs, as detailed in Appendix B.

### 3.2. Variational Reinforcement Fine-tuning

While our flow synthesis pipeline applies heuristic strategies to mitigate the inductive bias of experts, heuristically determining the golden flow for each sample remains challenging. To address this, we propose variational Reinforcement Fine-Tuning (RFT), which leverages a variational objective coupled with a *tailored reward* function and a *vicinal geometric shaping* scheme to ensure a better approximation of the target posterior $P_V$ by the policy $p_\theta(Z|X)$. Specifically, given the RL dataset $P_{\text{RFT}}(X, Y, E)$, let $R_\lambda(z_{0:i}) := \mathcal{F}(z_i) = \frac{R_\lambda(z_{0:i}\top)}{p_\theta(\top|z_{0:i})}$ be the reward of a trajectory ending at $z_i$, and $\top$ denotes the terminal state (*i.e.*, $\langle$/localize$\rangle$ token), we derive the objective for variational RFT by reformulating Equ. 1 (see Appendix A.2) as follows:

$$\mathcal{L}_{\text{vRFT}}(\theta) = \mathbb{E}_{\substack{X,Y,E \sim P_{\text{RFT}} \\ Z \sim p_\theta(\cdot|X)}} \left[ \sum_{0 \leq i \leq j \leq |Z|} \right. \tag{2}$$

$$\left. \left( \log \frac{R_\lambda(z_{0:i}\top) \prod_{k=i+1}^{j} p_\theta(z_k \mid z_{0:k-1}) p_\theta(\top \mid z_{0:j})}{R_\lambda(z_{0:j}\top) \, p_\theta(\top \mid z_{0:i})} \right)^2 \right]$$

Notably, Eq. 2 involves dense computations of rewards and transition probabilities for trajectories sharing the same sub-flow prefixes. For efficiency, we develop a parallel strategy to solve this objective, ensuring scalable optimization even for extensive perception chains (detailed in Appendix B.4).

**Reward Design.** To comprehensively characterize perceptual behaviors, given any sub-flow $z_{0:k} = (z_0, \ldots, z_k) \preceq Z$, we design a reward that jointly evaluates its *quality* and *reasoning efficacy* via a frozen reward model $p_\phi$ initialized from the PFlowNet. Formally, given an image $I$, consider an ideal caption distribution $p_{\phi,r_i}^+(\cdot) = p_\phi(\cdot \mid I_{r_i})$ and a noisy caption distribution $p_{\phi,r_i}^-(\cdot) = p_\phi(\cdot \mid I \setminus I_{r_i})$ for each perceptual state $z_i = \langle r_i, c_i \rangle \in z_{1:k}$, where $I_{r_i} = \text{Crop}(r_i, I)$ refers to the zoomed-in evidence targeted by $r_i$, while $I \setminus I_{r_i}$ denotes the less-informative portion outside $r_i$. We define the likelihood scores $p_\phi^+(z_i) := p_{\phi,r_i}^+(c_i)$, $p_\phi^-(z_i) := p_{\phi,r_i}^-(c_i)$, and induce the sample-level *quality* term from the expectation of *contrastive distillation* objective between the policy captions $q_{r_i}(\cdot) = p_\theta(\cdot \mid X, z_{<i}, r_i)$ and these two distributions:

$$\sum_{i=1}^{k} \left[ D_{\text{KL}}\Big(q_{r_i}(\cdot) \,\|\, p_{\phi,r_i}^-(\cdot)\Big) - D_{\text{KL}}\Big(q_{r_i}(\cdot) \,\|\, p_{\phi,r_i}^+(\cdot)\Big) \right]$$
$$= \sum_{i=1}^{k} \mathbb{E}_{c_i \sim q_{r_i}}[\log p_\phi^+(z_i) - \log p_\phi^-(z_i)] \rightsquigarrow \prod_{i=1}^{k} \frac{p_\phi^+(z_i)}{p_\phi^-(z_i)}.$$

Intuitively, maximizing this objective encourages each $q_{r_i}(\cdot)$ to be closer to the ideal distribution $p_{\phi, r_i}^+(\cdot)$, while moving it away from the noisy distribution $p_{\phi, r_i}^-(\cdot)$. This facilitates visually grounded and semantically specific captions, while suppressing generic descriptions induced by language priors or reward hacking. Furthermore, we measure the *reasoning efficacy* of the sampled flow $z_{0:k} \preceq Z$ by its sample-wise information gain for deriving the target response $Y$, characterized by the conditional log-likelihood improvement:

$$\log p_\phi(Y \mid z_{0:k}\top, X) - \log p_\phi(Y \mid X).$$

For any fixed data $(X, Y)$, the term $\log p_\phi(Y \mid X)$ is constant *w.r.t.* the flow; thus, maximizing $\log p_\phi(Y \mid z_{0:k}\top, X)$ favors perceptual flows with higher utility for inducing the target response $Y$. Finally, the overall reward is defined as

$$R(z_{0:k}\top) = \left(\prod_{i=1}^{k} \frac{p_\phi^+(z_i)}{p_\phi^-(z_i)}\right) p_\phi(Y \mid z_{0:k}\top, X). \quad (3)$$

**Vicinal Geometric Shaping.** While the designed reward $R(Z)$ characterizes the *utility* of a perceptual flow, it does not encode any geometric bias and may therefore encourage excessive exploration, yielding invalid trajectories outside the support $\mathcal{S}_V$. Motivated by *Vicinal Risk Minimization*, we introduce *vicinal geometric shaping* that constrains variational inference to a vicinity around the expert prior. Distinct from existing methods that enforce strict alignment between the policy and visual prior, our scheme targets only samples outside the vicinity, balancing sufficient exploration with validity to discover high-efficacy flows. We first define the *directed* Chamfer IoU between two RoI sets $A$ and $B$:

$$IoU_{A\to B} = \frac{1}{|A|} \sum_{a\in A} \sup_{b\in B} IoU(a, b), \ \ IoU(a, b) = \frac{a \cap b}{a \cup b},$$

and the *symmetrized* Chamfer-IoU distance:

$$d_{\text{IoU}}(A, B) = 1 - 0.5 * (IoU_{A\to B} + IoU_{B\to A}). \quad (4)$$

For any $(X, Y, E) \sim P_{\text{RFT}}$, we define the $\varepsilon$-vicinity of the reference $E$ by a ball $\mathcal{B}_\varepsilon(E) := \{z_{0:k} \mid d_{\text{IoU}}(r_{1:k}, E) \leq \varepsilon\}$ and introduce an energy weight $\omega_\lambda(z_{0:k}, E) := \exp\big(-\lambda \mathbb{I}(z_{0:k} \notin \mathcal{B}_\varepsilon(E))\big)$, where $\lambda$ is a hyper-parameter. Specifically, since $z_0$ is defined as the planning state without RoI, we define $\omega_\lambda(z_0, E) = 1$. We then shape the reward as

$$R_\lambda(z_{0:k}\top) := R(z_{0:k}\top)\,\omega_\lambda(z_{0:k}, E), \quad (5)$$

which penalizes excursions outside the vicinity and encourages $p_\theta$ to concentrate probability mass near $\mathcal{B}_\varepsilon(E)$.

### 3.3. Theoretical Analysis

In this section, we derive an *idealized* performance bound for PFlowNet under strict assumptions (see Appendix A.1) to characterize the effect of its key hyperparameters. By examining the limiting regimes of this bound, we show that the standard MLE and expert-guided RL arise as *special cases* of PFlowNet, establishing a *guaranteed improvement*.

Let $(X, Y) \sim P_{\text{RFT}}$ be any data tuple. We denote the expert annotation as $E \sim P(\cdot \mid X, Y)$ and the *golden* evidence as $G \sim P(\cdot \mid X, Y)$. Any perceptual flow is denoted by $Z := (z_0, \langle R, C\rangle)$, parametrized by the planning state $z_0$, predicted RoI $R$, and captions $C$. To formalize the relationship between $Z, E, G$, we define the *valid support* $\mathcal{S}_V$ and the *expert vicinity* $\mathcal{B}_\varepsilon$ based on $d_{\text{IoU}}$ with $\sigma, \varepsilon \in [0, 1]$:

$$\mathcal{S}_V := \{Z \mid d_{\text{IoU}}(R, G) \leq \sigma\}, \ \ \mathcal{B}_\varepsilon := \{Z \mid d_{\text{IoU}}(R, E) \leq \varepsilon\}.$$

Accordingly, denote $s_V$ and $s_\mathcal{B}$ as the probability masses associated with the support and the vicinity, respectively:

$$s_V := P(\mathcal{S}_V \mid X, Y), \quad s_\mathcal{B} := P(\mathcal{B}_\varepsilon \mid X, Y).$$

Thereby, we model the learning objective using a $\lambda$-shaped posterior distribution $P_\lambda(Z \mid X, Y, E)$. This formulation reweights the prior $P(Z \mid X, Y)$ to concentrate density around the expert vicinity via a shaping function $\omega_\lambda$:

$$P_\lambda(Z \mid X, Y, E) := P(Z \mid X, Y)\,\omega_\lambda(Z, E) / \mathcal{Z}_\lambda.$$

where the partition function $\mathcal{Z}_\lambda$ is given by

$$\mathcal{Z}_\lambda = \int P(Z \mid X, Y)\,\omega_\lambda(Z, E)\,dZ = s_\mathcal{B} + e^{-\lambda}(1 - s_\mathcal{B}).$$

Let $P_V(Z \mid X, Y) := P(Z \mid X, Y)/s_V$ be the target posterior for *idealized perceptual behaviors*. We now establish the *TV distance* bound between $p_\theta(Z \mid X)$ and this posterior $P_V$.

**Theorem 3.1** (Total Variation Distance Bound). *Under Assump. A.1, A.2, we suppose valid support $\mathcal{S}_V$ satisfies $d_{\text{eff}}$-regularity, where $d_{\text{eff}}$ is its effective dimension; thus, $\exists \kappa \geq 1$ such that $q := s_\mathcal{B}/s_V \geq \kappa(\varepsilon/\sigma)^{d_{\text{eff}}}$. Suppose the model $p_\theta$ is expressive and let $\theta^\star$ be the global minimizer of $\mathcal{L}_{\text{vRFT}}(\theta)$. The total variation distance between the policy $p_{\theta^\star}(Z \mid X)$ and the target posterior $P_V(Z \mid X, Y)$ is bounded by:*

$$D_{\text{TV}}(p_{\theta^\star}(\cdot \mid X), P_V(\cdot \mid X, Y)) \leq \frac{1}{2\,\mathcal{Z}_\lambda} \cdot$$
$$\left(q\,|s_V - \mathcal{Z}_\lambda| + (1 - q)\,|e^{-\lambda}s_V - \mathcal{Z}_\lambda| + e^{-\lambda}(1 - s_V)\right).$$

*Remark* 3.2 (Limit Analysis *w.r.t.* $\lambda$). As $\lambda \to 0$, the bound $D_{\text{TV}} \to (1 - s_V)$, dominated by the inherent sparsity of the valid support; this implies PFlowNet discards geometric constraints and *degrades to standard MLE*. Conversely, as $\lambda \to \infty$, the bound $D_{\text{TV}} \to (1 - q)$, where $q$ quantifies the discrepancy between the expert and golden priors. Thereby, the performance of PFlowNet is bottlenecked by expert bias, *i.e.*, *degenerating to expert-guided RLVR*.

*Remark* 3.3 (Limit Analysis *w.r.t.* $\varepsilon$). As $\varepsilon \to 0$, the vicinity contracts to a singularity ($q \to 0$); this forces the shaping energy to act indiscriminately on all trajectories, rendering the reward signal uninformative and ultimately loosening the bound. Conversely, increasing $\varepsilon$ within the valid region ($\mathcal{B}_\varepsilon \subseteq \mathcal{S}_V$) monotonically improves coverage ($q \uparrow$) and tightens the bound. However, if $\varepsilon$ exceeds the tolerance $\sigma$, the vicinity inevitably encompasses invalid regions, which *dilutes* the geometric guidance and degrades performance.

*Table 1.* Comparison with competitive alternatives on TreeBench (*left*) and MME-RealWorld-Lite (*right*).

| | | Perception | | | | | Reasoning | | | | | | Perception | | | | | Reasoning | | |
|---|---|---|---|---|---|---|---|---|---|---|---|---|---|---|---|---|---|---|---|---|
| | Overall | Attributes | Material | Phy. State | Obj. Retr. | OCR | Per. Trans. | Ordering | Con. & Oc. | Spa. Cont. | Comparison | Overall | OCR | Remote Sen. | Diag. & Tab. | Monitoring | Auto. Driv. | OCR | Diag. & Tab. | Monitoring | Auto. Driv. |
| *General Large Vision-Language Models* | | | | | | | | | | | | | | | | | | | | | |
| LLaVA-OV-7B | 37.3 | 55.2 | 53.8 | 56.5 | 50.0 | 32.4 | 21.2 | 22.8 | 41.5 | 72.4 | 36.4 | 43.7 | 80.0 | 40.0 | 56.0 | 31.7 | 39.4 | 65.0 | 33.0 | 38.0 | 32.0 |
| LLaVA-OV-72B | 40.5 | 62.1 | 53.8 | 65.2 | 62.3 | 36.8 | 12.9 | 28.1 | 53.7 | 65.5 | 47.7 | 48.7 | 79.2 | 50.7 | 67.0 | 37.9 | 40.0 | 76.0 | 41.0 | 38.7 | 39.3 |
| InternVL3-8B | 38.8 | 51.7 | 69.2 | 56.5 | 56.3 | 33.7 | 21.2 | 24.6 | 39.0 | 72.4 | 43.2 | 47.9 | 83.6 | 49.3 | 75.0 | 34.5 | 36.9 | 70.0 | 44.0 | 40.0 | 37.0 |
| InternVL3-38B | 42.0 | 51.7 | 61.5 | 52.2 | 68.8 | 51.5 | 12.9 | 33.3 | 56.1 | 65.5 | 38.6 | 51.0 | 85.6 | 56.0 | 71.0 | 42.6 | 40.0 | 77.0 | 45.0 | 47.3 | 35.0 |
| InternVL3-78B | 46.4 | 62.1 | 61.5 | 52.2 | 68.8 | 52.9 | 16.5 | 33.3 | 61.0 | 86.2 | 45.5 | 52.3 | 87.6 | 54.7 | 77.0 | 42.6 | 36.6 | 76.0 | 56.0 | 46.0 | 40.3 |
| Qwen2.5-VL-7B | 37.0 | 55.2 | 53.8 | 56.5 | 62.5 | 27.9 | 20.0 | 35.1 | 39.0 | 44.8 | 43.2 | 42.3 | 87.6 | 32.7 | 83.0 | 27.3 | 30.0 | 72.0 | 62.0 | 28.7 | 23.0 |
| Qwen2.5-VL-32B | 42.5 | 51.7 | 53.8 | 69.6 | 62.5 | 54.4 | 16.5 | 33.3 | 46.3 | 62.1 | 38.6 | 45.6 | 87.2 | 40.7 | 83.0 | 29.5 | 40.7 | 74.0 | 60.0 | 27.3 | 29.5 |
| Qwen2.5-VL-72B | 42.2 | 65.5 | 69.2 | 56.5 | 56.3 | 48.5 | 11.8 | 33.3 | 51.2 | 72.4 | 38.6 | 43.7 | 90.8 | 34.0 | 87.0 | 27.9 | 30.6 | 74.0 | 61.0 | 26.7 | 25.5 |
| Qwen3-VL-4B | 42.2 | 48.3 | 61.5 | 65.2 | 81.3 | 35.3 | 18.8 | 31.6 | 46.3 | 86.2 | 43.2 | 47.1 | 90.8 | 44.7 | 87.0 | 34.8 | 32.6 | 72.0 | 64.0 | 43.4 | 24.3 |
| Qwen3-VL-8B | 44.9 | 65.5 | 53.9 | 65.2 | 75.0 | 64.7 | 12.9 | 24.6 | 48.8 | 72.4 | 43.2 | 48.6 | 92.8 | 57.3 | 87.0 | 36.4 | 31.4 | 73.0 | 70.0 | 39.3 | 25.3 |
| Qwen3-VL-32B | 45.2 | 60.3 | 63.4 | 58.1 | 83.6 | 30.3 | 24.2 | 39.7 | 47.7 | 85.2 | 51.4 | 52.0 | 91.6 | 47.3 | 96.0 | 36.1 | 42.9 | 76.0 | 77.0 | 42.7 | 30.0 |
| *Visually Grounded Reasoning Models* | | | | | | | | | | | | | | | | | | | | | |
| Pixel-Reasoner | 39.0 | 58.6 | 61.5 | 65.2 | 50.0 | 48.5 | 14.1 | 31.6 | 39.0 | 44.8 | 40.9 | 49.7 | 89.6 | 52.0 | 86.0 | 38.9 | 30.9 | 71.0 | 72.0 | 46.0 | 32.5 |
| DeepEyes | 37.5 | 62.1 | 53.8 | 65.2 | 68.8 | 51.5 | 11.8 | 24.6 | 36.6 | 51.7 | 47.7 | 53.2 | 90.0 | 52.7 | 89.0 | 43.3 | 33.4 | 76.0 | 69.0 | 44.0 | 35.0 |
| DeepEyesV2 | 40.7 | 65.5 | 69.2 | 56.5 | 62.5 | 55.9 | 11.8 | 35.1 | 46.3 | 37.9 | 36.4 | 52.4 | 85.6 | 49.3 | 89.0 | 45.8 | 33.4 | 70.0 | 76.0 | 44.0 | 37.0 |
| Thyme | 38.2 | 48.2 | 46.1 | 69.5 | 50.0 | 51.4 | 22.3 | 21.0 | 41.3 | 44.8 | 34.0 | 54.4 | 90.4 | 56.7 | 86.0 | 46.3 | 38.5 | 78.0 | 71.0 | 48.0 | 36.0 |
| TreeVGR | 50.4 | 65.5 | 53.8 | 82.6 | 68.8 | 63.3 | 22.4 | 36.8 | 61.0 | 69.0 | 45.5 | 54.9 | 87.6 | 50.7 | 83.0 | 47.0 | 43.4 | 74.0 | 66.0 | 51.3 | 39.0 |
| **PFlowNet** (Ours) | 55.3 | 65.5 | 69.2 | 80.2 | 75.0 | 77.9 | 20.0 | 40.4 | 56.1 | 82.8 | 56.8 | 67.0 | 95.6 | 69.3 | 90.0 | 53.6 | 58.2 | 83.0 | 76.0 | 70.0 | 53.5 |
| Δ *v.s.* Base Model | ↑10.4 | – | ↑15.3 | ↑15.0 | – | ↑13.2 | ↑7.1 | ↑15.8 | ↑7.3 | ↑10.4 | ↑13.6 | ↑18.4 | ↑2.8 | ↑12.0 | ↑3.0 | ↑17.2 | ↑26.8 | ↑10.0 | ↑6.0 | ↑30.7 | ↑28.2 |

**Theorem 3.4** (Guaranteed Improvement over Baselines)**.** *Let* $D_{\text{TV}}(\lambda, \varepsilon)$ *be the TV bound in Theorem 3.1. For any* $\varepsilon$ *satisfying* $\mathcal{B}_\varepsilon \subseteq \mathcal{S}_{\text{V}}$, *there exists an intensity* $\lambda^\star$ *such that*

$$D_{\text{TV}}(\lambda^\star, \varepsilon) \leq \min\{1 - s_{\text{V}}, 1 - q\}.$$

*For fixed* $\lambda = \lambda^\star$, *the bound is strictly decreasing in* $q$ ($\varepsilon \uparrow$).

**Remark** 3.5. This confirms that with proper calibration of intensity $\lambda$ and radius $\varepsilon$, PFlowNet strictly tightens the *idealized* TV bound of standard MLE and expert-guided RLVR.

*Proof.* Refer to Appendix A.4 for the proofs. □

## 4. Experiment

We initialize PFlowNet from Qwen3-VL-8B and evaluate it against representative baselines spanning both general-purpose and fine-grained visual tasks. More implementation details and experimental setups are provided in App. B; C.

### 4.1. Main Results

**General-purpose Tasks.** As shown in Table 1, PFlowNet exhibits robust capabilities in both perception and reasoning. It delivers substantial gains over the base model across all scenarios, achieving overall improvements of 10.4% on TreeBench and 18.4% on MME-RealWorld-Lite. Notably, driven by our reasoning-oriented reward design, these gains are particularly pronounced on reasoning-heavy subsets. Furthermore, PFlowNet outperforms both grounded RLVR-based methods, *e.g.*, TreeVGR (Wang et al., 2025a), Pixel Reasoner (Su et al., 2025), and agentic frameworks, *e.g.*, DeepEyes (Zheng et al., 2025b), Thyme (Zhang et al., 2025). It yields the best average performance, surpassing the nearest competitors by 5.3% and 12.6% on TreeBench and MME-RealWorld-Lite, respectively; and sets SOTA records on 89% of sub-tasks, underscoring its generalization.

**Fine-grained Visual Understanding.** As presented in Table 2, PFlowNet achieves SOTA results across all benchmarks, outperforming both representative baselines and general LVLMs. Notably, although the Qwen3-VL series incorporates architectural improvements (*e.g.*, DeepStack) to enhance fine-grained capabilities, PFlowNet still delivers clear gains of 13%, 8%/8.8%, and 2.5%–7% on V, HR-Bench (4k/8k), and ScreenSpot, respectively. These improvements are primarily concentrated in reasoning-oriented subsets, such as spatial reasoning and cross-objective relationship recognition. This validates our key insight: PFlowNet yields high-utility perceptual results that enhance visual reasoning while ensuring reliability. Consequently, despite being built on Qwen3-VL-8B, PFlowNet matches the performance of the larger Qwen3-VL-32B on these challenging tasks.

### 4.2. In-depth Analysis

**Performance-Efficiency Trade-off.** As shown in Figure 5, PFlowNet exhibits an excellent balance between performance and efficiency. Compared to the agentic frameworks, PFlowNet substitutes complex tool or code executions with carefully designed structured perceptual flows to efficiently encode visual thoughts. This results in significantly shorter context lengths and reduced inference latency without compromising performance. In contrast to TreeVGR, PFlowNet decouples the process into flow generation and flow-guided visual reasoning, which incurs affordable computational costs to substantially improve perceptual quality and utility, thereby significantly boosting visual reasoning performance.

**Test-Time Scaling.** Theoretically, PFlowNet's variational objective (Equ. 2) ensures a diverse rationale distribution, whereas grounded RLVR often implicitly optimizes a highly sharp distribution due to rigid alignment with sparse ex-

*Table 2.* Performance comparison on fine-grained visual tasks: visual search, high-resolution VQA, and GUI grounding.

| | V* Bench | | | HR-Bench 4K | | | HR-Bench 8K | | |
|---|---|---|---|---|---|---|---|---|---|
| | *Overall* | *Attribute* | *Spatial* | *Overall* | *Single* | *Cross* | *Overall* | *Single* | *Cross* |
| *General Large Vision-Language Models* | | | | | | | | | |
| GPT-4o-1120 (OpenAI, 2024) | 66.0 | – | – | 59.0 | 70.0 | 48.0 | 55.5 | 62.0 | 49.0 |
| LLaVA-OV-72B (Li et al., 2024a) | 73.8 | 80.9 | 63.2 | 66.3 | 76.5 | 56.0 | 60.9 | 68.8 | 53.0 |
| InternVL3-8B (Zhu et al., 2025) | 72.3 | 73.0 | 71.1 | 70.8 | 79.3 | 62.3 | 62.0 | 64.3 | 59.8 |
| InternVL3-38B | 77.5 | 77.4 | 77.6 | 76.3 | 83.5 | 69.0 | 67.0 | 71.3 | 62.8 |
| Qwen2.5-VL-7B (Bai et al., 2025b) | 74.3 | 77.4 | 69.7 | 72.1 | 88.8 | 55.5 | 68.8 | 83.5 | 54.0 |
| Qwen2.5-VL-32B | 85.9 | 83.5 | 89.5 | 74.8 | 89.3 | 60.3 | 71.6 | 86.5 | 56.8 |
| Qwen2.5VL-72B | 84.8 | 90.8 | 80.9 | 79.4 | 88.8 | 70.0 | 76.3 | 84.3 | 68.3 |
| Qwen3-VL-4B (Bai et al., 2025a) | 74.9 | 78.3 | 69.7 | 73.5 | 84.8 | 62.3 | 67.1 | 83.5 | 50.7 |
| Qwen3-VL-8B | 77.5 | 80.2 | 73.7 | 72.4 | 88.5 | 56.3 | 68.1 | 82.0 | 54.3 |
| Qwen3-VL-32B | 87.4 | 87.0 | 88.2 | 82.1 | 94.0 | 70.2 | 74.8 | 90.1 | 59.5 |
| *Visually Grounded Reasoning Models* | | | | | | | | | |
| PixelReasoner (Su et al., 2025) | 80.6 | 83.5 | 76.3 | 72.9 | 86.0 | 60.3 | 66.9 | 80.0 | 54.3 |
| DeepEyes (Zheng et al., 2025b) | 90.0 | 92.1 | 86.8 | 75.1 | 91.3 | 59.0 | 72.6 | 86.8 | 58.5 |
| DeepEyesV2 (Hong et al., 2025) | 81.8 | 81.7 | 80.3 | 77.9 | 92.8 | 63.0 | 73.8 | 88.5 | 59.0 |
| Thyme (Zhang et al., 2025) | 82.2 | 83.5 | 80.3 | 77.0 | 91.0 | 63.0 | 72.0 | 86.5 | 57.5 |
| TreeVGR (Wang et al., 2025a) | 87.4 | 89.5 | 84.2 | 77.1 | 89.5 | 64.8 | 72.8 | 86.0 | 59.5 |
| **PFlowNet (Ours)** | 90.6 | 91.4 | 89.5 | 80.4 | 91.2 | 69.5 | 76.9 | 89.0 | 64.8 |
| Δ *v.s.* Base Model | ↑13 | ↑11 | ↑16 | ↑8.0 | ↑2.7 | ↑13.2 | ↑8.8 | ↑7.0 | ↑10.5 |

| | ScreenSpot | |
|---|---|---|
| | *v2* | *Pro* |
| *General LVLMs* | | |
| GPT-4o-1120 (OpenAI, 2024) | 18.1 | 0.8 |
| Claude Comp. Use (Hu et al., 2024a) | - | 17.1 |
| OpenAI CUA (OpenAI, 2025b) | 87.9 | 23.4 |
| Qwen2-VL-7B | - | 1.6 |
| Qwen2.5-VL-3B (Bai et al., 2025b) | 68.4 | 23.9 |
| Qwen2.5-VL-7B | 73.6 | 29.0 |
| Qwen2.5-VL-72B | 87.1 | 43.6 |
| Qwen3-VL-8B (Bai et al., 2025a) | 92.7 | 54.6 |
| Kimi-VL-16B-MoE (Team et al., 2025) | 92.8 | 34.5 |
| SeedVL-1.5 (Guo et al., 2025a) | 95.0 | 60.9 |
| *GUI Grounding Models* | | |
| SeeClick (Cheng et al., 2024) | 55.1 | 1.1 |
| OS-Atlas-4B (Wu et al., 2024) | 71.9 | 3.7 |
| OS-Atlas-7B | 84.1 | 18.9 |
| UI-TARS-2B (Qin et al., 2025) | 84.7 | 27.7 |
| ViGoRL-7B (Sarch et al., 2025) | 86.5 | 31.1 |
| **PFlowNet (Ours)** | 95.1 | 61.8 |
| Δ *v.s.* Base Model | ↑2.4 | ↑7.2 |

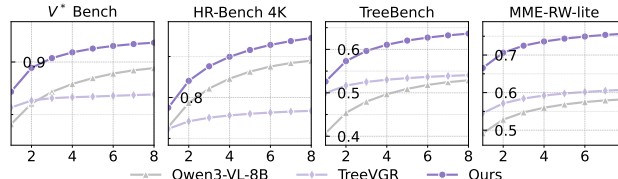

*Figure 5.* Performance-efficiency trade-offs of Qwen3-VL 8B, PFlowNet, TreeVGR, DeepEyesV2, and Thyme. All models are evaluated on an H200 via VLMEvalKit (see App. C). Latency (s) denotes the average inference time per sample, and context (k) is the averaged token-level length. Notably, a smaller occupied region *below the shared axis* indicates lower computational (temp. & spatial) cost.

*Figure 6.* Pass@$k$ curves (*i.e.*, $k \in [1,8]$) of different methods across both fine-grained and general-purpose benchmarks.

*Table 3.* Ablation study on the training recipe, reward design and flow-guided reasoning, where $\pm$ and $+$ denote $P^+/P^-$ and $P^+$.

| | | Inference | | Rewards | | Treebench | | V* | MME-RW |
|---|---|---|---|---|---|---|---|---|---|
| | | $Z$ | $I_{\text{RoI}}$ | $R_{\text{cl}}$ | $R_{\text{gain}}$ | Acc | mIoU | Acc | Acc |
| (1) | Base Model | - | - | - | - | 44.9 | - | 77.5 | 46.0 |
| (2) | + SFT | ✓ | ✓ | - | - | 48.3 | 44.2 | 83.7 | 54.2 |
| (3) | | ✓ | ✓ | $\pm$ | - | 51.5 | 43.7 | 85.3 | 59.5 |
| (4) | + RFT | ✓ | ✓ | - | ✓ | 52.8 | 40.5 | 87.4 | 62.8 |
| (5) | | ✓ | ✓ | $\pm$ | ✓ | 55.3 | 38.2 | 90.6 | 67.0 |
| (6) | | ✓ | ✓ | $+$ | ✓ | 52.2 | 36.4 | 88.1 | 65.5 |
| (7) | PFlowNet | ✓ | - | $\pm$ | ✓ | 54.5 | 38.2 | 89.4 | 66.4 |
| (8) | PFlowNet | - | ✓ | $\pm$ | ✓ | 49.2 | 38.2 | 83.8 | 52.1 |

pert trajectories. Empirical analysis in Figure 6 validates this: while TreeVGR achieves high Pass@1 accuracy, it yields negligible performance gains as the computational budget ($k$) increases, particularly in challenging scenarios (*e.g.*, V*, TreeBench). This phenomenon aligns with recent findings by (Yue et al., 2025). Notably, by incorporating variational inference with tailored reward design and geometric shaping, PFlowNet achieves superior Pass@1 results while demonstrating robust test-time scaling capabilities.

**Case Study.** Figure 7 qualitatively highlights PFlowNet's superior reliability. Different from TreeVGR, where coupled perception-reasoning often yields geometrically precise yet semantically misaligned boxes due to sparse reward signals, PFlowNet utilizes dense contrastive rewards to enforce strict visual-textual dependency ($c_i$ on $r_i$), ensuring faithful interpretability. Interestingly, we observe that PFlowNet often

prioritizes precise localization and then expands its visual scope. We attribute this to the shaping energy $\omega_\lambda(z_{0:k}, E)$ derived from the sequence-level metric $d_{\text{IoU}}$. Specifically, when $k$ is small, the scarcity of participating RoIs compels the model to maximize the precision of each individual proposal; however, this constraint naturally relaxes as the sequence elongates, facilitating comprehensive reasoning.

### 4.3. Ablation Studies

**Reward & Macro Design.** Table 3 validates the effectiveness of SFT and the synthetic flows, while the proposed RFT strategy yields further substantial gains (1, 2, 6). Further-

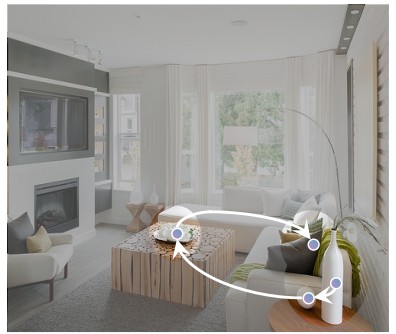

TreeVGR: `<think>` The question asks if the sculpture to the left of the vase looks white and small. First, identify the vase `<box>`[800, 720, 864, 997]`</box>` which is white and small-sized. Then, find the sculpture to its left `<box>`[293, 613, 395, 700]`</box>`. It is also white and small-sized. So, the sculpture to the left of the vase looks white and small.`</think>` `<answer>` Yes `</answer>`

Qwen3-VL 8B Instruct: `<thinking>` The sculpture to the left of the vase does not look white and small. The sculpture appears to be a light-colored, possibly wooden or stone, and it is larger than the vase. `</thinking>` `<answer>` No, the sculpture is large. `</answer>`

PFlowNet: `<analyze>` The question asks whether the sculpture located to the left of the vase is both white and small. To answer this, I need to identify the vase and then locate the sculpture immediately to its left. I will then examine the color and relative size of that sculpture compared to surrounding objects in the image. `</analyze>` `<localize>` `<box>`[792, 677, 885, 977]`</box>` This is the white vase positioned on the side table. *The sculpture is near this vase. This is the white, spherical sculpture located on the wooden side table to the right of the sofa, next to the vase. I should further examine the image.* `<box>`[385, 672, 488, 729]`</box>` This is a white, bowl-shaped sculpture placed on the wooden coffee table. It is located to the left of the vase on the side table, but not directly adjacent. `<box>`[712, 677, 867, 819]`</box>` This is the gray cushion on the sofa, which seems not relevant to the question. I have fully explored the image. `</localize>`

`<thinking>` *Upon re-examining the image and supplied visual evidence,* I confirm that there is a white, spherical sculpture located on the wooden side table to the right of the sofa, next to the tall white vase. Therefore, the sculpture to the left of the vase is indeed white and small. `</thinking>` `<answer>` Yes, the sculpture to the left of the vase is white and small. `</answer>`

*Question: Does the sculpture to the left of the vase look white and small?* **Ground Truth**: *Yes.*

*Figure 7.* Qualitative comparison of visual reasoning across different methods. PFlowNet enables precise yet comprehensive exploration of visual evidence and effectively anchors the reasoning process to perceptual outcomes, producing the most reliable and accurate answers.

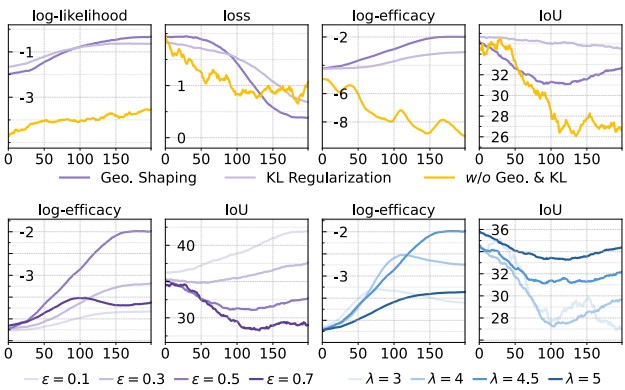

*Figure 8.* Ablation study on the geometric shaping scheme.

*Table 4.* Cross-scale improvements of PFlowNet across general-purpose visual tasks and fine-grained understanding.

| | V* Bench | HRB 4k | HRB 8k | TreeBench | MME-RW | CVB 2D | MMB | AI2D | ChartQA |
|---|---|---|---|---|---|---|---|---|---|
| Qwen3-VL 8B | 77.5 | 76.1 | 72.7 | 44.9 | 48.6 | 79.5 | 85.3 | 84.0 | 82.6 |
| Qwen3-VL 4B | 74.8 | 73.5 | 67.1 | 42.2 | 47.1 | 78.2 | 84.5 | 83.8 | 82.1 |
| w SFT | 79.1 | 75.9 | 69.3 | 44.1 | 50.6 | 79.8 | 85.7 | 84.9 | 83.5 |
| Δ v.s. Base Model | ↑4.3 | ↑2.4 | ↑2.2 | ↑1.9 | ↑3.5 | ↑1.6 | ↑1.2 | ↑1.1 | ↑1.4 |
| w RFT | 83.5 | 77.9 | 70.9 | 46.8 | 55.3 | 82.0 | 87.2 | 90.0 | 84.5 |
| Δ v.s. Base Model | ↑8.7 | ↑4.4 | ↑3.8 | ↑4.6 | ↑8.2 | ↑3.8 | ↑2.7 | ↑2.7 | ↑2.3 |
| Qwen3-VL 32B | 87.4 | 82.1 | 74.8 | 45.2 | 52.0 | 81.5 | 87.7 | 89.0 | 83.1 |
| w SFT | 89.1 | 83.5 | 76.3 | 46.7 | 54.3 | 82.8 | 88.4 | 89.9 | 84.2 |
| Δ v.s. Base Model | ↑1.7 | ↑1.4 | ↑1.3 | ↑1.4 | ↑2.3 | ↑1.3 | ↑0.7 | ↑0.9 | ↑1.0 |
| w RFT | 91.6 | 85.5 | 77.5 | 49.2 | 58.8 | 85.2 | 89.8 | 91.5 | 85.7 |
| Δ v.s. Base Model | ↑4.2 | ↑3.1 | ↑2.8 | ↑4.0 | ↑6.8 | ↑3.7 | ↑2.1 | ↑2.5 | ↑2.6 |

more, the quality and efficacy rewards exhibit a collaborative effect during the RFT, while the contrastive formulation also plays a crucial role (3–6). Regarding the macro design, we examine the impact of input information during the reasoning stage (7, 8). Interestingly, incorporating external fine-grained visual features yields only marginal improvements; in contrast, removing the perceptual flow leads to severe performance degradation. This result indicates that the perceptual flow functions as more than a localization tool; it serves as a critical explicit semantic anchor. By translating visual thoughts into a structured textual prefix, the flow effectively conditions the LVLM's autoregressive generation, thereby bridging the semantic gap and guiding the reasoning trajectory more directly than raw visual features.

**Geometric Shaping.** As illustrated in Figure 8, the geometric shaping stabilizes training and permits broader exploration than typical KL regularization, effectively mitigating SFT inductive bias. The initial drop and subsequent resurgence in $d_{\text{IoU}}$ reflect a healthy transition from early exploration to late exploitation, yielding reliable yet high-efficacy perceptual behaviors. Further ablations of vicinal radius $\varepsilon$ and shaping intensity $\lambda$ show trends consistent with our analysis (Theorem 3.1): excessive radius $\varepsilon$ (*e.g.*, 0.7) may lead to over-exploration into invalid supports, while a too small radius $\varepsilon$ enforces expert bias, preventing reasoning-oriented exploration. Likewise, small intensity $\lambda$ causes substantial training instability, whereas large $\lambda$ restricts exploration.

**Cross-Scale Evaluations.** While our primary experiments utilize the Qwen3-VL 8B backbone to maintain parameter parity with baselines, we extend our evaluation to the Qwen3-VL 4B and 32B variants to verify the scalability of PFlowNet. We employ the consistent training data and pipeline detailed in Section B, with specific adjustments for computational efficiency: *the SFT and RFT phases are restricted to 1 and 2 epochs*, respectively, with an SFT global batch size of 128. As shown in Table 4, even under this computationally efficient training regime, backbones across different scales derive clear margins of improvement from our framework. Specifically, across general-purpose and fine-grained visual understanding tasks, our SFT stage yields average performance gains of 2.2% and 1.3% for Qwen3-VL 4B and Qwen3-VL 32B, respectively. Building upon this, our tailored RFT strategy further delivers substantial improvements exceeding 2.3%, effectively demonstrating the scalability of our approach.

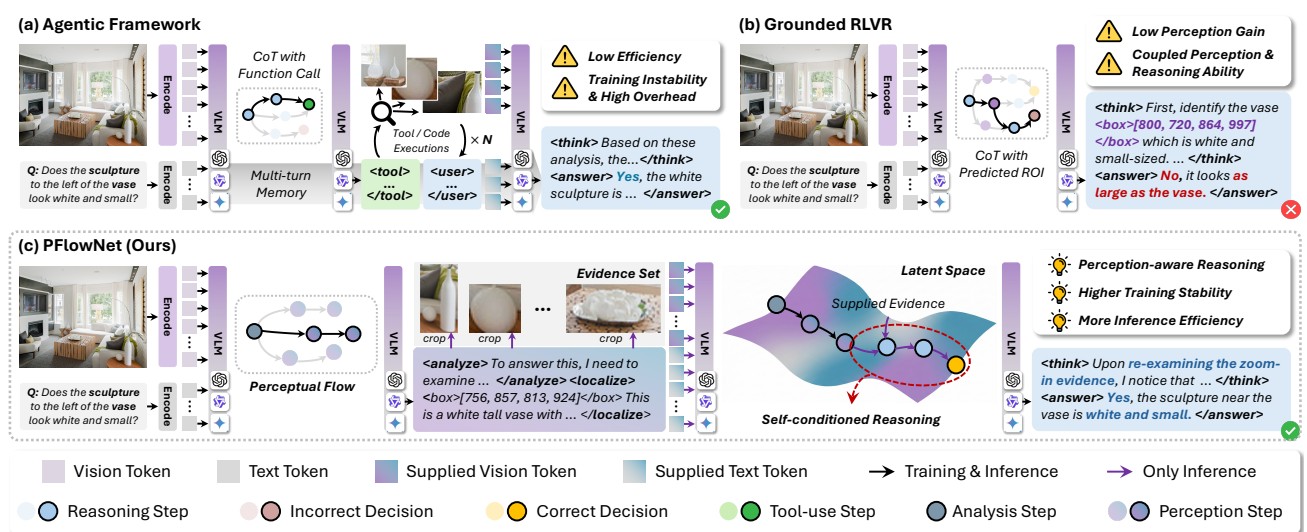

*Figure 9.* Framework comparisons between different paradigms, where (a) agentic frameworks rely on multi-turn tool executions for a "perceive-then-reason" process, (b) grounded RLVR integrates perception into reasoning in a single turn, and (c) the proposed PFlowNet decouples perception from reasoning via a two-stage perceptual flow, achieving robust yet efficient visually grounded reasoning.

## 5. Related Work

**Agentic Frameworks** equip LVLMs with dynamic image manipulation capabilities via multi-turn tool/code executions, thereby facilitating reliable visual reasoning, as shown in Figure 9(a). Specifically, several works explore the potential of enhancing visual reasoning by integrating external sandbox tools (Zheng et al., 2025b; Hong et al., 2025; Su et al., 2025; Liu et al., 2025c). Thyme (Zhang et al., 2025) realizes "*thinking with images*" by enabling the model to generate and execute code, while Visual Sketchpad (Hu et al., 2024b) empowers MLLMs with a sketching workspace to extend CoT with intermediate visual thoughts. Similarly, VaCoT (Xu et al., 2025) leverages visual tools to mitigate performance degradation on low-quality inputs. Despite these advances, CodeDance (Song et al., 2025) highlights a critical trade-off between tool utilization and intrinsic reasoning capabilities. Moreover, due to the entanglement of perception with complex invocations, these methods often suffer from excessive context and high latency. Different from these methods, the proposed PFlowNet utilizes structured perceptual flows as the proxy of perceptual behaviors and efficiently achieves comparable high-quality visual reasoning based on the self-conditioned generation framework with supplied fine-grained features, as shown in Figure 9(c).

**Grounded RLVR** maximizes the geometric consistency between intermediate visual rationales and external expert prior via reinforcement learning to regularize the reasoning process, as presented in Figure 9(b). Prior works (Wang et al., 2025a;b; Liu et al., 2025a;b; Wei et al., 2026) represent perception via normalized *bounding box coordinates* interleaved with stepwise reasoning, optimizing the policy by maximizing the Intersection over Union (IoU) with the ground truth. In contrast, ViGoRL (Sarch et al., 2025) and GUI-R1 (Luo et al., 2025) employ a point-based perceptual proxy, encouraging more precise perception through spatial distance constraints. Moreover, MIRG-RL (Zheng et al., 2025a) explores Grounded RLVR in multi-image scenarios. However, our experiment results reveal the risk of rigidly enforcing alignment with biased and sparse expert priors. Furthermore, these methods often overlook the semantic coherence between visual rationales and the surrounding textual context, potentially compromising reasoning performance. These limitations are effectively mitigated by the multi-dimensional reward function in PFlowNet.

## 6. Conclusion

This paper introduces PFlowNet, a novel framework based on structured perceptual flows that enables high-quality and interpretable visual reasoning. PFlowNet incorporates a carefully designed reinforcement fine-tuning strategy, comprising a tailored reward function with vicinal geometric shaping, which allows LVLMs to explore reasoning-oriented yet valid perceptual behaviors. Formal analysis establishes a provable performance guarantee for PFlowNet. Extensive experiments further demonstrate its superiority across both general-purpose and fine-grained tasks. Notably, the empirical analysis highlights PFlowNet's excellent performance-efficiency balance and robust test-time scaling property.

## Acknowledgment

We gratefully thank the computational support jointly provided by Inclusion AI, Ant Group and the Infrastructure Platform Department of Shanghai AI Laboratory. The work

was additionally supported from the National Natural Science Foundation of China (62476092), the Natural Science Foundation of Chongqing (CSTB2024NSCQ-MSX0877), and the Science and Technology Commission of Shanghai Municipality (24JD1401300), and the AI project from the economic and information commission of Shanghai (Grant No. 2024-GZL-RGZN-01038), and Fundamental Research Funds for the Central Universities, and ECNU Multifunctional Platform for Innovation (001).

## Impact Statement

This paper introduces PFlowNet to enhance visual reasoning in LVLMs. Ethically, the model may inherit biases from its pre-trained backbones and the synthetic data used for training. There is also a potential risk of misuse in surveillance or automated tasks. In terms of societal impact, PFlowNet improves interpretability and reduces hallucinations, fostering safer deployment in complex visual environments.

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

# A. Omitted Technical Details

**Roadmap.** *We organize the theoretical analysis as follows. In **Appendix A.1**, we formalize the probabilistic framework and preliminary definitions. **Appendix A.2** provides the rigorous derivation of our variational objective, stemming from the general Sub-Trajectory Balance principle. Building on this, we introduce necessary regularity assumptions to facilitate tractable analysis and establish two auxiliary lemmas in **Appendix A.3**: **Lemma A.3** demonstrates that the shaped reward is proportional to an exponentially tilted posterior $P_\lambda$, i.e., $R(Z) \propto P_\lambda$, while **Lemma A.4** proves that the global optimum of the policy recovers this tilted distribution, i.e., $p_{\theta^\star}(Z \mid X) \propto R(Z) \propto P_\lambda$. Finally, in **Appendix A.4**, we present the complete proofs of the main theorems. By bridging the optimal policy $p_{\theta^\star}$ and the target valid posterior $P_V$ via the tilted distribution $P_\lambda$, we derive the Total Variation (TV) bound in **Theorem 3.1**. We conclude with an algebraic analysis of this bound in **Theorem 3.4**, confirming that PFlowNet provides strictly tighter guarantees compared to limiting baselines.*

## A.1. Preliminaries

*Basic variables and flow notation.* Let $(X, Y, E) \sim P_{\mathrm{RFT}}$ denote a sample from the RFT dataset, where $X \coloneqq \langle I, T \rangle$ consists of an image $I$ and an instruction $T$, $Y = (y_1, \ldots, y_L)$ is the response sequence generated by the verifier when given the synthetic flow $Z_{\mathrm{s}}(X, E)$, and $E$ is a reference set of RoIs (*e.g.*, expert evidence) used for vicinal geometry shaping. For theoretical analysis, we assume an intractable target joint distribution

$$P(X, Y, Z, \top),$$

defined over inputs $X$, outputs $Y$, latent perceptual flows $Z$, and a terminal symbol $\top$ that marks the end of a flow. Following Definition 2.2, a perceptual flow is written as a finite trajectory

$$Z = (z_0 \to z_1 \to \cdots \to z_K), \qquad z_k = \langle r_k, c_k \rangle \quad (k \geq 1),$$

where $z_0$ is the planning state, and each perceptual state $z_k$ consists of a RoI $r_k$ and its caption $c_k$. We write

$$r_{1:K} \coloneqq (r_1, \ldots, r_K), \qquad c_{1:K} \coloneqq (c_1, \ldots, c_K).$$

Throughout the analysis, we condition on a fixed flow length $K$ for each input instance, while allowing the geometric precision and traversal order of the RoIs to vary across latent realizations. For mathematical convenience, RoIs are represented by normalized coordinates $r_k \in [0,1]^4$, *e.g.*, $r_k = (x_1, y_1, x_2, y_2)$ with $0 \leq x_1 < x_2 \leq 1$ and $0 \leq y_1 < y_2 \leq 1$. All geometric distances below are defined on this normalized coordinate system. For any $k \in \{0, 1, \ldots, K\}$, define the prefix or sub-flow

$$z_{0:k} \coloneqq (z_0 \to z_1 \to \cdots \to z_k),$$

and use $z_{0:k}\top$ to denote its terminated prefix.

*Model factorization.* Given an input $X$, let

$$\mathcal{R}(X) = \{R_1(X), \ldots, R_M(X)\}$$

denote the support of all admissible unordered RoI bags appearing in valid flows under the target process $P(Z \mid X)$. Each $R_j(X) = \{r_i^j\}_{i=1}^K$ is a set of RoIs with cardinality $K$ and represents one admissible unordered realization of the intrinsic visual evidence. An ordered tuple $r_{1:K}$ is a permutation of a bag $R$, denoted by $r_{1:K} \in \mathbb{S}_R$, where $\mathbb{S}_R$ is the permutation class induced by the elements of $R$.

PFlowNet parameterizes a variational distribution $p_\theta(Z \mid X)$ over flows and a conditional generator $p_\theta(Y \mid X, Z)$. We do not assume a specific architecture; it suffices that $p_\theta(Z \mid X)$ induces a forward transition kernel over states. For notational brevity, we use the generic autoregressive factorization

$$p_\theta(Z \mid X) = p_\theta(z_0 \mid X) \prod_{k=1}^K p_\theta(z_k \mid z_{0:k-1}, X), \tag{A1-1}$$

and define forward/backward sub-trajectory probabilities as in Eq. (1).

*Structural hypotheses.* We impose three hypotheses used in the subsequent analysis. First, the planning state is assumed to be uniquely determined by the input $X$: there exists a deterministic mapping $g$ such that

$$z_0 = g(X), \qquad P(z_0 \mid \cdot, X) = \mathbf{1}\{z_0 = g(X)\}. \tag{A1-2}$$

Second, for each RoI $r$, define a deterministic crop operator

$$I_r \ = \ \mathrm{Crop}(r, I), \qquad I_{r_{j:k}} := (I_{r_j}, \ldots, I_{r_k}).$$

The captioning likelihood is assumed to depend on $I$ only through the cropped visual content:

$$P(c_{j:k} \mid I, r_{j:k}) \ = \ P(c_{j:k} \mid I_{r_{j:k}}). \tag{A1-3}$$

We also use $I \setminus I_r$ to denote the complement context outside the RoI $r$. Third, since the contrastive caption term applies only to perceptual states $z_{k \geq 1}$, we set the boundary condition at the planning state as

$$\frac{P^+(z_0)}{P^-(z_0)} \ := \ 1, \qquad \text{s.t.} \qquad R(z_0 \top) = P(Y \mid z_0 \top, X). \tag{A1-4}$$

*Vicinal support.* Assume there exists a conceptual golden RoI trajectory $G_R = (g_1, \ldots, g_K)$ that captures the intrinsic visual evidence for $X$. For a tolerance parameter $\sigma \in [0, 1]$, define the support of valid flows as

$$\mathcal{S}_{\mathrm{V}} \ := \ \Big\{ Z : d_{\mathrm{IoU}}\big(\{r_1, \ldots, r_K\}, \{g_1, \ldots, g_K\}\big) \leq \sigma \Big\},$$

which is assumed to be nonempty. Its posterior mass is denoted by

$$s_{\mathrm{V}} \ := \ P(\mathcal{S}_{\mathrm{V}} \mid X, Y). \tag{A1-5}$$

Given a reference RoI set $E$ and radius $\varepsilon \in [0, 1]$, define the $\varepsilon$-vicinity of $E$ by

$$\mathcal{B}_\varepsilon(E) \ := \ \Big\{ z_{0:k} : d_{\mathrm{IoU}}\big(\{r_1, \ldots, r_k\}, E\big) \leq \varepsilon \Big\}.$$

Given a complete flow $Z \in \mathcal{B}_\varepsilon(E)$, its terminal RoI set $\{r_1, \ldots, r_K\}$ lies within the $\varepsilon$-vicinity of $E$. We define

$$s_{\mathcal{B}} \ := \ P(\mathcal{B}_\varepsilon(E) \mid X, Y), \qquad q \ := \ \frac{s_{\mathcal{B}}}{s_{\mathrm{V}}}.$$

We assume $\varepsilon$ is sufficiently small such that

$$\mathcal{B}_\varepsilon(E) \subseteq \mathcal{S}_{\mathrm{V}}, \qquad \text{hence} \qquad 0 \leq s_{\mathcal{B}} \leq s_{\mathrm{V}} \leq 1. \tag{A1-6}$$

*Reward-induced tilted posterior.* For any $\lambda \geq 0$, the shaping weight $\omega_\lambda$ induces an exponentially tilted posterior

$$P_\lambda(Z \mid X, Y, E) \ := \ \frac{P(Z \mid X, Y)\, \omega_\lambda(Z, E)}{\mathcal{Z}_\lambda}, \qquad \mathcal{Z}_\lambda \ := \ \int P(Z \mid X, Y)\, \omega_\lambda(Z, E)\, dZ. \tag{A1-7}$$

Since $\omega_\lambda(Z, E) = 1$ on $\mathcal{B}_\varepsilon(E)$ and $\omega_\lambda(Z, E) = e^{-\lambda}$ on its complement, the normalizer admits the closed form

$$
\begin{aligned}
\mathcal{Z}_\lambda &= \int_{\mathcal{B}_\varepsilon(E)} P(Z \mid X, Y)\, dZ + e^{-\lambda} \int_{\mathcal{B}_\varepsilon(E)^c} P(Z \mid X, Y)\, dZ \\
&= P(\mathcal{B}_\varepsilon(E) \mid X, Y) + e^{-\lambda} P(\mathcal{B}_\varepsilon(E)^c \mid X, Y) \\
&= s_{\mathcal{B}} + e^{-\lambda}(1 - s_{\mathcal{B}}).
\end{aligned}
$$

Using $s_{\mathcal{B}} = q\, s_{\mathrm{V}}$, we equivalently have

$$\mathcal{Z}_\lambda = q\, s_{\mathrm{V}} + e^{-\lambda}\big(1 - q\, s_{\mathrm{V}}\big). \tag{A1-8}$$

Finally, define the valid-support posterior, which serves as the *target distribution* in total-variation distance:

$$P_{\mathrm{V}}(Z \mid X, Y) \ := \ P(Z \mid X, Y, Z \in \mathcal{S}_{\mathrm{V}}) \ = \ \frac{P(Z \mid X, Y)\, \mathbf{1}\{Z \in \mathcal{S}_{\mathrm{V}}\}}{s_{\mathrm{V}}}. \tag{A1-9}$$

## A.2. Derivation of Variational Objective

In this section, we provide the detailed algebraic derivation connecting the general Sub-Trajectory Balance (SubTB) objective in Equation (1) to our specific variational loss function in Equation (1). We begin with the squared log-difference form of the SubTB objective for a sub-trajectory $z_{i:j}$ (where $0 \leq i < j \leq K$), which is formulated as

$$\mathcal{L}(z_{i:j}) = \left( \log \frac{\mathcal{F}(z_i) \cdot \mathcal{T}_F(z_{i:j})}{\mathcal{F}(z_j) \cdot \mathcal{T}_B(z_{j:i})} \right)^2. \tag{A2-1}$$

Based on the structural constraints of Perceptual Flow (Definition 2.2) and the variational distribution $p_\theta$, we instantiate the terms in Equation (A2-1) as follows. Given the tree-structured autoregressive generation, the backward path from any state is deterministic, effectively collapsing the backward transition to unity, *i.e.*, $\mathcal{T}_B(z_{j:i}) = 1$. Conversely, the forward transition $\mathcal{T}_F(z_{i:j})$ is governed by the product of conditional likelihoods parameterized by $\theta$, yielding $\mathcal{T}_F(z_{i:j}) = \prod_{k=i+1}^{j} p_\theta(z_k \mid z_{0:k-1})$. Furthermore, we formulate the scalar flow $\mathcal{F}(z_k)$ at state $z_k$ as the terminal reward normalized by the termination policy, formally defined as $\mathcal{F}(z_k) := R_\lambda(z_{0:k}\top)/p_\theta(\top \mid z_{0:k})$.

Substituting the above definitions into the ratio term of Equation (A2-1), denoted as $\Delta_{i,j}$, we have

$$\begin{aligned}
\Delta_{i,j} &= \log \left( \frac{\mathcal{F}(z_i) \cdot \mathcal{T}_F(z_{i:j})}{\mathcal{F}(z_j) \cdot 1} \right) \\
&= \log \left( \frac{\left( \frac{R_\lambda(z_{0:i}\top)}{p_\theta(\top \mid z_{0:i})} \right) \cdot \prod_{k=i+1}^{j} p_\theta(z_k \mid z_{0:k-1})}{\left( \frac{R_\lambda(z_{0:j}\top)}{p_\theta(\top \mid z_{0:j})} \right)} \right) \\
&= \log \left( \left( \frac{R_\lambda(z_{0:i}\top)}{p_\theta(\top \mid z_{0:i})} \right) \cdot \prod_{k=i+1}^{j} p_\theta(z_k \mid z_{0:k-1}) \cdot \left( \frac{p_\theta(\top \mid z_{0:j})}{R_\lambda(z_{0:j}\top)} \right) \right) \\
&= \log \left( \frac{R_\lambda(z_{0:i}\top) \cdot \prod_{k=i+1}^{j} p_\theta(z_k \mid z_{0:k-1}) \cdot p_\theta(\top \mid z_{0:j})}{R_\lambda(z_{0:j}\top) \cdot p_\theta(\top \mid z_{0:i})} \right).
\end{aligned}$$

Then, given a trajectory $Z = (z_0, \ldots, z_K)$, we derive the loss by summing the squared log-ratios over all valid sub-flow $z_{i:j}$

$$\mathcal{L}_{\text{vRFT}}(Z, \theta) = \sum_{0 \leq i \leq j \leq K} \Delta_{i,j}^2 = \sum_{0 \leq i \leq j \leq K} \left( \log \frac{R_\lambda(z_{0:i}\top) \prod_{k=i+1}^{j} p_\theta(z_k \mid z_{0:k-1}) \, p_\theta(\top \mid z_{0:j})}{R_\lambda(z_{0:j}\top) \, p_\theta(\top \mid z_{0:i})} \right)^2. \tag{A2-2}$$

Finally, we extend this formulation to the empirical data setting $(X, Y, E) \sim P_{\text{data}}$, where $E = \{e_l\}_{l=1}^{L} \subset \mathbb{N}^4$ denotes the set of expert-annotated Regions of Interest (RoIs). Consequently, the objective in Equation (1) is given by

$$\mathcal{L}_{\text{vRFT}}(\theta) = \mathbb{E}_{\substack{X,Y,E \sim P_{\text{RFT}} \\ Z \sim p_\theta(\mathcal{Z} \mid X)}} \left[ \sum_{0 \leq i \leq j \leq |Z|} \left( \log \frac{R_\lambda(z_{0:i}\top) \prod_{k=i+1}^{j} p_\theta(z_k \mid z_{0:k-1}) \, p_\theta(\top \mid z_{0:j})}{R_\lambda(z_{0:j}\top) \, p_\theta(\top \mid z_{0:i})} \right)^2 \right]. \tag{A2-3}$$

where $Z$ denotes a sampled perceptual flow. In practice, we approximate the expectation over $Z \sim p_\theta(\mathcal{Z} \mid X)$ using Monte Carlo estimation with $L = 8$ *i.i.d.* sampled flows per input. The optimized policy is given by $\theta^\star = \arg\min_\theta \mathcal{L}_{\text{vRFT}}(\theta)$.

## A.3. Assumptions and Auxiliary Lemmas

**Assumption A.1** (Uniform Prior). Motivated by the empirical results, we adopt a uniform prior of *admissible* RoI space $\mathcal{R}(X)$. Formally, given $X$, for all $Z(z_0, R, C) \sim P(Z \mid X)$ where $R \in \mathcal{R}(X)$ and $|R| = K$, we have

$$P(R \mid X) = \frac{1}{|\mathcal{R}(X)|}, \quad P(r_{1:K} \mid R, X) = \frac{1}{K!},$$

where $r_{1:K} \in \mathbb{S}_R$ and $\mathbb{S}_R$ is the symmetric group of $\forall r \in R$.

**Assumption A.2** (Faithful Captioning with Non-Informative Prior). We assume $\forall c \in C$ is faithful when conditioned on the cropped evidence $I_r$, and become non-informative when the evidence is absent (*i.e.*, $I \setminus I_r$). Formally, let $\mathcal{C}$ be a finite space containing *all* candidate captions $c$, we have

$$P(c_{j:k} \mid I_{r_{j:k}}) = \prod_{i=j}^{k} P(c_i \mid I_{r_i}), \quad P(c \mid I \setminus I_r) = \frac{1}{|\mathcal{C}|}.$$

**Lemma A.3** (Reward Consistency). *Under Assumptions A.1 and A.2, for all $(X, Y, E) \sim P_{\text{RFT}}$, the shaped multi-dimensional reward $R_\lambda(Z)$ satisfy*

$$R_\lambda(Z) := R_\lambda(z_{0:k}\top) \propto P_\lambda(Z \mid X, Y, E).$$

*Proof.* Leveraging the definition of multi-dimensional reward in Equations (3) and (5), given any sub-flow $z_{0:k}$, we have

$$R_\lambda(z_{0:k}\top) := \left( \prod_{i=1}^{k} \frac{P(c_i \mid I_{r_i})}{P(c_i \mid I \setminus I_{r_i})} \right) P(Y \mid z_{0:k}\top, X) \, \omega_\lambda(z_{0:k}, E).$$

Invoking Assumption A.2, for all $i \in \{1, \dots, K\}$, we have $P(c_i \mid I \setminus I_{r_i}) = 1/|\mathcal{C}|$ where $\mathcal{C}$ is a finite space containing all candidate captions $c$. Plugging this into the above yields

$$R_\lambda(Z) = \left( \prod_{i=1}^{K} P(c_i \mid I_{r_i}) \right) |\mathcal{C}|^K \, P(Y \mid Z, X) \, \omega_\lambda(Z, E),$$

$$\propto \left( \prod_{i=1}^{K} P(c_i \mid I_{r_i}) \right) P(Y \mid Z, X) \, \omega_\lambda(Z, E),$$
(A3-1)

where the proportionality absorbs the factor $|\mathcal{C}|^K$, which only depends on $X$. By the factorization of Assump. A.2, we have

$$\prod_{i=1}^{K} P(c_i \mid I_{r_i}) = P(c_{1:K} \mid I_{r_{1:K}}).$$

Subsequently, Equation (A3-1) becomes

$$\boxed{R_\lambda(Z) \propto P(c_{1:K} \mid I_{r_{1:K}}) \, P(Y \mid Z, X) \, \omega_\lambda(Z, E).}$$
(A3-2)

For the standard posterior, by Bayes' rule, we have

$$P(Z \mid X, Y) = \frac{P(Y, Z \mid X)}{P(Y \mid X)},$$

and $P(Y \mid X)$ is a normalizing constant independent of $Z$. Thus,

$$P(Z \mid X, Y) \propto P(Y, Z \mid X) = P(Z \mid X) \, P(Y \mid Z, X) \equiv P(R, z_0, r_{1:K}, c_{1:K} \mid X) \, P(Y \mid Z, X),$$

$$\propto \underbrace{P(R \mid X) \, P(z_0 \mid R, X) \, P(r_{1:K} \mid z_0, R, X) \, P(c_{1:K} \mid r_{1:K}, z_0, R, X)}_{P(R, z_0, r_{1:K}, c_{1:K} \mid X)} \, P(Y \mid Z, X).$$
(A3-3)

Based on the structural constraint that the traversal order is a uniform permutation over the support set (Assumption A.1), the specific sequence $r_{1:K}$ is conditionally independent of the deterministic planning state $z_0$ given $X$, yielding $P(r_{1:K} \mid z_0, R, X) = P(r_{1:K} \mid R, X)$. Leveraging the deterministic crop operator $\text{Crop} : \langle I, r \rangle \mapsto I_r$, we have

$$P(c_{1:K} \mid r_{1:K}, z_0, R, X) = P(c_{1:K} \mid r_{1:K}, R, I, T),$$
$$= P(c_{1:K} \mid I_{r_{1:K}}, z_0, R, T),$$
$$= P(c_{1:K} \mid \cdot, I_{r_{1:K}}).$$

By Assumption A.2 and Equation (A1-3), the caption $c$ is only conditioned on the attended region $I_r$; thus, we have

$$P(c_{1:K} \mid r_{1:K}, z_0, R, X) = P(c_{1:K} \mid I_{r_{1:K}}).$$

Therefore, substituting the above into the Equation (A3-3), the standard posterior $P(Z \mid X, Y)$ satisfies

$$P(Z \mid X, Y) \propto \underbrace{P(R \mid X)}_{\text{Support Prior}} \underbrace{P(z_0 \mid \cdot, X)}_{\text{Deterministic State}} \underbrace{P(r_{1:K} \mid R, X)}_{\text{Ordering Prior}} \underbrace{P(c_{1:K} \mid I_{r_{1:K}})}_{\text{Caption Sequence}} \underbrace{P(Y \mid Z, X)}_{\text{Perceptual Efficacy}} .$$

By leveraging Assumption A.1 (*i.e.*, uniform support and ordering prior) and Equation (A1-2), we have

$$
\begin{aligned}
P(Z \mid X, Y) \; &\propto \; \frac{1}{|\mathcal{R}(X)|K\,!} \; P(c_{1:K} \mid I_{r_{1:K}}) \, P(Y \mid Z, X), \\
&\propto \; P(c_{1:K} \mid I_{r_{1:K}}) \, P(Y \mid Z, X).
\end{aligned}
\tag{A3-4}
$$

Based on Equations (A1-7) and (A3-4), we have

$$
\boxed{
\begin{aligned}
P_\lambda(Z \mid X, Y, E) \; &\propto \; P(Z \mid X, Y) \, \omega_\lambda(Z, E), \\
&\propto \; P(c_{1:K} \mid I_{r_{1:K}}) \, P(Y \mid Z, X) \, \omega_\lambda(Z, E).
\end{aligned}
}
\tag{A3-5}
$$

Consequently, combining Equation (A3-2) with Equation (A3-5) gives

$$
R_\lambda(Z) \; \propto \; P_\lambda(Z \mid X, Y, E).
$$

This completes the proof. $\qquad\square$

**Lemma A.4** (Posterior Matching Induced by the Variational Objective). Under Assumption A.1 and A.2, suppose the policy $p_\theta$ is expressive and $\theta^\star$ globally minimizes $\mathcal{L}_{\mathrm{vRFT}}(\theta)$, for every $(X, Y, E) \sim P_{\mathrm{RFT}}$ we have

$$
p_{\theta^\star}(Z \mid X) \; \propto \; R_\lambda(Z) \; \propto \; P_\lambda(Z \mid X, Y, E).
$$

*Proof.* Based on Equation (A1-1), the forward trajectory probability induced by policy $p_\theta$ is defined as

$$
p_\theta(Z, \top \mid X) \; \coloneqq \; \Big( \prod_{k=1}^{K} p_\theta(z_k \mid z_{0:k-1}) \Big) p_\theta(\top \mid z_{0:K}).
\tag{A3-6}
$$

Based on Equation (A2-2), when policy $p_\theta$ is expressive enough and the optimization reaches a solution with $\mathcal{L}_{\mathrm{VRFT}}(Z, \theta) = 0$ for all valid $(i, j)$, we have each squared term $\Delta_{i,j}$ equals 0. Formally, when for every $0 \le i \le j \le K$, the following holds

$$
\log \frac{R_\lambda(z_{0:i}\top) \prod_{k=i+1}^{j} p_{\theta^\star}(z_k \mid z_{0:k-1}) \, p_{\theta^\star}(\top \mid z_{0:j})}{R_\lambda(z_{0:j}\top) \, p_{\theta^\star}(\top \mid z_{0:i})} \; = \; 0.
$$

Exponentiating the above yields the exact balance constraints

$$
R_\lambda(z_{0:i}\top) \Big( \prod_{k=i+1}^{j} p_{\theta^\star}(z_k \mid z_{0:k-1}) \Big) p_{\theta^\star}(\top \mid z_{0:j}) \; = \; R_\lambda(z_{0:j}\top) \, p_{\theta^\star}(\top \mid z_{0:i}).
$$

Taking $(i, j) = (0, K)$ in the previous equation gives

$$
R_\lambda(z_0\top) \underbrace{\Big( \prod_{k=1}^{K} p_{\theta^\star}(z_k \mid z_{0:k-1}) \Big) p_{\theta^\star}(\top \mid z_{0:K})}_{p_{\theta^\star}(Z, \top \mid X)} \; = \; R_\lambda(z_{0:K}\top) \, p_{\theta^\star}(\top \mid z_0).
\tag{A3-7}
$$

The product term in (A3-7) is precisely the forward trajectory probability $p_{\theta^\star}(Z, \top \mid X)$ in (A3-6), hence

$$
p_{\theta^\star}(Z, \top \mid X) \; = \; \frac{p_{\theta^\star}(\top \mid z_0)}{R_\lambda(z_0\top)} \, R_\lambda(z_{0:K}\top).
\tag{A3-8}
$$

Based on Equations (A1-2) and (A1-4), $z_0$ is uniquely determined by $X$; thus, $R_\lambda(z_0\top)$, *i.e.*, $P(Y \mid z_0\top, X)$, is a normalization choice at the boundary. Hence, the prefactor $\frac{p_{\theta^\star}(\top \mid z_0)}{R_\lambda(z_0\top)}$ does not depend on the particular trajectory realization $Z$. Therefore, Equation (A3-8) implies the proportionality

$$
p_{\theta^\star}(Z \mid X) \; \coloneqq \; p_{\theta^\star}(Z, \top \mid X) \; \propto \; R_\lambda(Z).
$$

*i.e.*, minimizing the variational loss $\mathcal{L}_{\mathrm{VRFT}}(Z, \theta)$ yields an optimum policy $p_{\theta^\star}(Z \mid X)$ that is proportional to the reward $R_\lambda(Z)$. Thereby, based on the conclusion of **Lemma A.3**, *i.e.*, $R_\lambda(z) \propto P_\lambda(Z \mid X, Y, E)$, we have

$$
\boxed{p_{\theta^\star}(Z \mid X) \; \propto \; P_\lambda(Z \mid X, Y, E).}
\tag{A3-9}
$$

This completes the proof. $\qquad\square$

## A.4. Proofs

**Restatement of Theorem 3.1** [Variation Distance Bound]. *Under Assumption A.1, A.2, we suppose the valid support $\mathcal{S}_V$ satisfies $d_{\text{eff}}$-regularity, where $d_{\text{eff}}$ is its effective dimension; thus, $\exists \kappa \geq 1$ such that $q := s_\mathcal{B}/s_V \geq \kappa(\varepsilon/\sigma)^{d_{\text{eff}}}$. Suppose the model $p_\theta$ is expressive and let $\theta^\star$ be the global minimizer of $\mathcal{L}_{\text{vRFT}}(\theta)$. The total variation distance between the policy $p_{\theta^\star}(Z \mid X)$ and the target posterior $P_V(Z \mid X, Y)$ is bounded by:*

$$D_{\text{TV}}(p_{\theta^\star}(\cdot \mid X), P_V(\cdot \mid X, Y)) \leq \frac{1}{2\,\mathcal{Z}_\lambda} \cdot \left( q\,|s_V - \mathcal{Z}_\lambda| + (1-q)\,|e^{-\lambda}s_V - \mathcal{Z}_\lambda| + e^{-\lambda}(1 - s_V) \right).$$

*Remark* A.5 (Limit Analysis *w.r.t.* $\lambda$). We analyze the asymptotic behavior of the bound by evaluating the limits of $\lambda$

$\diamondsuit$ *MLE Regime* ($\lambda \to 0$): Since $e^{-\lambda} \to 1$, the partition function $\mathcal{Z}_\lambda \to s_\mathcal{B} + (1 - s_\mathcal{B}) = 1$. The bound simplifies to $\frac{1}{2}\left(q(1 - s_V) + (1 - q)(1 - s_V) + (1 - s_V)\right) = 1 - s_V$. Dominated by inherent data variance ($s_V$), PFlowNet discards geometric constraints and *degrades to standard MLE*.

$\diamondsuit$ *RLVR Regime* ($\lambda \to \infty$): Since $e^{-\lambda} \to 0$, we have $\mathcal{Z}_\lambda \to s_\mathcal{B}$. The bound becomes $\frac{1}{2s_\mathcal{B}}\left(q(s_V - s_\mathcal{B}) + (1 - q)s_\mathcal{B} + 0\right)$. Using $s_\mathcal{B} = qs_V$, the numerator simplifies to $qs_V(1 - q) + s_Vq(1 - q) = 2s_\mathcal{B}(1 - q)$, yielding a final bound of $1 - q$. Here, performance is bottlenecked by the expert bias ($q$), *degenerating to expert-guided RLVR*.

*Remark* A.6 (Limit Analysis *w.r.t.* $\varepsilon$). The vicinity radius $\varepsilon$ affects the bound through the coverage ratio $q$. As $\varepsilon \to 0$, the vicinity contracts to a singularity, implying $q \to 0$ and $s_\mathcal{B} \to 0$. Consequently, $\mathcal{Z}_\lambda \to e^{-\lambda}$. Substituting these into Theorem 3.1, the bound approaches $\frac{1}{2e^{-\lambda}}\left(0 + |e^{-\lambda}s_V - e^{-\lambda}| + e^{-\lambda}(1 - s_V)\right) = 1 - s_V$. This algebraic equivalence to the MLE bound confirms that as the reward signal becomes uninformative, the geometric guidance vanishes. Conversely, increasing $\varepsilon$ (where $\mathcal{B}_\varepsilon \subseteq \mathcal{S}_V$) increases $q$, monotonically tightening the bound towards $(1 - q)$. However, if $\varepsilon > \sigma$, the vicinity encompasses invalid regions, diluting the guidance and degrading performance.

*Proof.* Since $\theta^\star$ acts as a global minimizer of the objective $\mathcal{L}_{\text{vRFT}}$, which is formulated as an expectation over $(X, Y, E) \sim P_{\text{RFT}}$ (A2-3), the optimality condition derived in **Lemma A.4** holds for $P_{\text{RFT}}$-almost every data tuple. Thereby, we have:

$$p_{\theta^\star}(Z \mid X) \propto P_\lambda(Z \mid X, Y, E),$$

Since both sides are valid probability distributions normalized over the space of $Z$, this implies strict point-wise equality

$$p_{\theta^\star}(Z \mid X) = P_\lambda(Z \mid X, Y, E). \tag{A4-1}$$

*Remark*. While a realizable parametric model $p_\theta(Z \mid X)$ cannot analytically depend on $Y, E$, this equality characterizes the ideal behavior of the policy at the global optimum of the variational objective for a given training instance.

Then, recall the definition of total variation distance $D_{\text{TV}}$:

$$D_{\text{TV}}(P, Q) = \sup_{\mathcal{A}} |P(\mathcal{A}) - Q(\mathcal{A})| = \frac{1}{2}\int |p(z) - q(z)|\,dz, \tag{A4-2}$$

where $p, q$ are densities w.r.t. a common base measure. Based on Equation (A4-1), we will bound $D_{\text{TV}}(P_\lambda(\cdot \mid X, Y, E), P_V(\cdot \mid X, Y))$. Since $\mathcal{B}_\varepsilon(E) \subseteq \mathcal{S}_V$ (A1-6), we partition the flow space $\Omega$, *i.e.*, the support of $P(Z \mid X, Y)$, into three *disjoint* measurable regions:

$$\Omega = \underbrace{\mathcal{B}_\varepsilon(E)}_{\text{vicinal}} \sqcup \underbrace{\left(\mathcal{S}_V \setminus \mathcal{B}_\varepsilon(E)\right)}_{\text{valid}} \sqcup \underbrace{\mathcal{S}_V^c}_{\text{invalid}}. \tag{A4-3}$$

Based on the definition of the energy weight $\omega_\lambda$, we have

$$\omega_\lambda(z_{0:k}, E) := \exp\left(-\lambda \cdot \mathbb{I}\{z_{0:k} \notin \mathcal{B}_\varepsilon(E)\}\right) \quad \text{and} \quad \omega_\lambda(z_0, E) \equiv 1.$$

Thereby, leveraging Equation (A1-7), for any $Z$:

$\diamondsuit$ If $Z \in \mathcal{B}_\varepsilon(E)$, then $\omega_\lambda(Z, E) = 1$, hence

$$P_\lambda(Z \mid X, Y, E) = \frac{P(Z \mid X, Y)}{\mathcal{Z}_\lambda}. \tag{A4-4}$$

$\diamond$ If $Z \notin \mathcal{B}_\varepsilon(E)$, then $\omega_\lambda(Z, E) = e^{-\lambda}$, hence

$$P_\lambda(Z \mid X, Y, E) = \frac{e^{-\lambda} P(Z \mid X, Y)}{\mathcal{Z}_\lambda}. \tag{A4-5}$$

Leveraging Equation (A1-9), for the *target* posterior:

$\diamond$ If $Z \in \mathcal{S}_\mathrm{V}$, then

$$P_\mathrm{V}(Z \mid X, Y) = \frac{P(Z \mid X, Y)}{s_\mathrm{V}}. \tag{A4-6}$$

$\diamond$ If $Z \notin \mathcal{S}_\mathrm{V}$, then

$$P_\mathrm{V}(Z \mid X, Y) = 0. \tag{A4-7}$$

Based on Equations (A4-2) and (A4-3), we have

$$\begin{aligned}
2\, D_\mathrm{TV}(P_\lambda(Z \mid X, Y, E), P_\mathrm{V}(Z \mid X, Y)) = &\int_{\mathcal{B}_\varepsilon(E)} \left| P_\lambda(Z \mid X, Y, E) - P_\mathrm{V}(Z \mid X, Y) \right| dZ \\
&+ \int_{\mathcal{S}_\mathrm{V} \setminus \mathcal{B}_\varepsilon(E)} \left| P_\lambda(Z \mid X, Y, E) - P_\mathrm{V}(Z \mid X, Y) \right| dZ \\
&+ \int_{\mathcal{S}_\mathrm{V}^c} \left| P_\lambda(Z \mid X, Y, E) - P_\mathrm{V}(Z \mid X, Y) \right| dZ.
\end{aligned} \tag{A4-8}$$

We then evaluate the three integrals separately.

For $Z \in \mathcal{B}_\varepsilon(E) \subseteq \mathcal{S}_\mathrm{V}$, substituting Equations (A4-4) and (A4-6) into the first term in Equation (A4-8), we have

$$\begin{aligned}
\int_{\mathcal{B}_\varepsilon(E)} \left| \frac{P(Z|X,Y)}{\mathcal{Z}_\lambda} - \frac{P(Z|X,Y)}{s_\mathrm{V}} \right| dZ &= \int_{\mathcal{B}_\varepsilon(E)} P(Z \mid X, Y) \left| \frac{1}{\mathcal{Z}_\lambda} - \frac{1}{s_\mathrm{V}} \right| dZ \\
&= \left| \frac{1}{\mathcal{Z}_\lambda} - \frac{1}{s_\mathrm{V}} \right| \int_{\mathcal{B}_\varepsilon(E)} P(Z \mid X, Y)\, dZ \\
&= \left| \frac{1}{\mathcal{Z}_\lambda} - \frac{1}{s_\mathrm{V}} \right| s_\mathcal{B} \\
&= \frac{|s_\mathrm{V} - \mathcal{Z}_\lambda|}{\mathcal{Z}_\lambda\, s_\mathrm{V}}\, s_\mathcal{B}.
\end{aligned} \tag{A4-9}$$

Using $s_\mathcal{B} = q\, s_\mathrm{V}$ gives

$$\int_{\mathcal{B}_\varepsilon(E)} |P_\lambda(Z \mid X, Y, E) - P_\mathrm{V}(Z \mid X, Y)| = \frac{|s_\mathrm{V} - \mathcal{Z}_\lambda|}{\mathcal{Z}_\lambda}\, q.$$

For $Z \in \mathcal{S}_\mathrm{V} \setminus \mathcal{B}_\varepsilon(E)$, substituting Equations (A4-5) and (A4-6) into the second term in Equation (A4-8), we have

$$\begin{aligned}
\int_{\mathcal{S}_\mathrm{V} \setminus \mathcal{B}_\varepsilon(E)} \left| \frac{e^{-\lambda} P(Z|X,Y)}{\mathcal{Z}_\lambda} - \frac{P(Z|X,Y)}{s_\mathrm{V}} \right| dZ &= \int_{\mathcal{S}_\mathrm{V} \setminus \mathcal{B}_\varepsilon(E)} P(Z \mid X, Y) \left| \frac{e^{-\lambda}}{\mathcal{Z}_\lambda} - \frac{1}{s_\mathrm{V}} \right| dZ \\
&= \left| \frac{e^{-\lambda}}{\mathcal{Z}_\lambda} - \frac{1}{s_\mathrm{V}} \right| \int_{\mathcal{S}_\mathrm{V} \setminus \mathcal{B}_\varepsilon(E)} P(Z \mid X, Y)\, dZ \\
&= \left| \frac{e^{-\lambda}}{\mathcal{Z}_\lambda} - \frac{1}{s_\mathrm{V}} \right| (s_\mathrm{V} - s_\mathcal{B}) \\
&= \frac{|e^{-\lambda} s_\mathrm{V} - \mathcal{Z}_\lambda|}{\mathcal{Z}_\lambda\, s_\mathrm{V}} (s_\mathrm{V} - s_\mathcal{B}).
\end{aligned} \tag{A4-10}$$

Using $s_\mathcal{B} = q\, s_\mathrm{V}$, *i.e.*, $s_\mathrm{V} - s_\mathcal{B} = s_\mathrm{V}(1 - q)$ yields

$$\int_{\mathcal{S}_\mathrm{V} \setminus \mathcal{B}_\varepsilon(E)} |P_\lambda(Z \mid X, Y, E) - P_\mathrm{V}(Z \mid X, Y)| = \frac{|e^{-\lambda} s_\mathrm{V} - \mathcal{Z}_\lambda|}{\mathcal{Z}_\lambda} (1 - q).$$

For $Z \in \mathcal{S}_V^c$, we have $P_V(Z \mid X, Y) = 0$. Also $Z \in \mathcal{S}_V^c$ implies $Z \notin \mathcal{B}_\varepsilon(E)$ because $\mathcal{B}_\varepsilon(E) \subseteq \mathcal{S}_V$. Hence on $\mathcal{S}_V^c$, $P_\lambda(Z \mid X, Y, E) = \frac{e^{-\lambda} P(Z \mid X, Y)}{\mathcal{Z}_\lambda}$. Therefore, the second term in Equation (A4-8) becomes

$$
\begin{aligned}
\int_{\mathcal{S}_V^c} \left| P_\lambda(Z \mid X, Y, E) - P_V(Z \mid X, Y) \right| dZ &= \int_{\mathcal{S}_V^c} P_\lambda(Z \mid X, Y, E) \, dZ \\
&= \int_{\mathcal{S}_V^c} \frac{e^{-\lambda} P(Z \mid X, Y)}{\mathcal{Z}_\lambda} \, dZ \\
&= \frac{e^{-\lambda}}{\mathcal{Z}_\lambda} \int_{\mathcal{S}_V^c} P(Z \mid X, Y) \, dZ \qquad \text{(A4-11)} \\
&= \frac{e^{-\lambda}}{\mathcal{Z}_\lambda} P(\mathcal{S}_V^c \mid X, Y) \\
&= \frac{e^{-\lambda}}{\mathcal{Z}_\lambda} (1 - s_V).
\end{aligned}
$$

Combining Equations (A4-8) to (A4-11), the bound is given by

$$
\begin{aligned}
D_{\mathrm{TV}}(P_\lambda(Z \mid X, Y, E), P_V(Z \mid X, Y)) &= \frac{1}{2} \left( \frac{|s_V - \mathcal{Z}_\lambda|}{\mathcal{Z}_\lambda} q + \frac{|e^{-\lambda} s_V - \mathcal{Z}_\lambda|}{\mathcal{Z}_\lambda} (1 - q) + \frac{e^{-\lambda}}{\mathcal{Z}_\lambda} (1 - s_V) \right) \\
&= \frac{1}{2 \mathcal{Z}_\lambda} \left( q |s_V - \mathcal{Z}_\lambda| + (1 - q) |e^{-\lambda} s_V - \mathcal{Z}_\lambda| + e^{-\lambda} (1 - s_V) \right).
\end{aligned}
$$

Finally, substituting $p_{\theta^\star}(Z \mid X) = P_\lambda(Z \mid X, Y, E)$ yields the claimed result:

$$
\begin{aligned}
D_{\mathrm{TV}}\big(p_{\theta^\star}(\cdot \mid X), P_V(\cdot \mid X, Y)\big) &\leq D_{\mathrm{TV}}(P_\lambda(Z \mid X, Y, E), P_V(Z \mid X, Y)), \\
&\leq \frac{1}{2 \mathcal{Z}_\lambda} \left( q |s_V - \mathcal{Z}_\lambda| + (1 - q) |e^{-\lambda} s_V - \mathcal{Z}_\lambda| + e^{-\lambda} (1 - s_V) \right),
\end{aligned}
$$

where $\mathcal{Z}_\lambda, s_V$ are separately given by Equations (A1-8) and (A1-5).

This completes the proof. $\qquad\qquad\square$

**Restatement of Theorem 3.4** [Guaranteed Improvement over Baselines]. *Let $D_{\mathrm{TV}}(\lambda, \varepsilon)$ be the TV bound in Theorem 3.1. For any $\varepsilon$ satisfying $\mathcal{B}_\varepsilon \subseteq \mathcal{S}_V$, there exist an intensity $\lambda$ such that*

$$
D_{\mathrm{TV}}(\lambda^\star, \varepsilon) \ \leq \ \min\{\, 1 - s_V, \, 1 - q \,\}.
$$

*For fixed $\lambda = \lambda^\star$, the bound is strictly decreasing in $q$ ($\varepsilon \uparrow$).*

*Remark* A.7. This confirms that with proper calibration of intensity $\lambda$ and radius $\varepsilon$, PFlowNet strictly tightens the *idealized* TV bound of standard MLE and expert-guided RLVR.

*Proof.* Recall

$$
D_{\mathrm{TV}}(\lambda, \varepsilon) \ = \ \frac{1}{2 \mathcal{Z}_\lambda} \left( q |s_V - \mathcal{Z}_\lambda| + (1 - q) |e^{-\lambda} s_V - \mathcal{Z}_\lambda| + e^{-\lambda} (1 - s_V) \right), \qquad \mathcal{Z}_\lambda = s_\mathcal{B} + e^{-\lambda} (1 - s_\mathcal{B}), \quad \text{(A4-12)}
$$

where $q := s_\mathcal{B} / s_V$ and we assume the valid regime $\mathcal{B}_\varepsilon \subseteq \mathcal{S}_V$, hence $0 \leq s_\mathcal{B} \leq s_V \leq 1$ and $q \in [0, 1]$.

**Limiting baselines.** Let $\alpha := e^{-\lambda} \in (0, 1]$, so $\mathcal{Z}_\lambda = s_\mathcal{B} + \alpha(1 - s_\mathcal{B})$. As $\lambda \to 0$ we have $\alpha \to 1$ and thus $\mathcal{Z}_\lambda \to 1$. Substituting into $D_{\mathrm{TV}}$ yields

$$
\lim_{\lambda \to 0} D_{\mathrm{TV}}(\lambda, \varepsilon) = \frac{1}{2} \left( q |s_V - 1| + (1 - q)|s_V - 1| + (1 - s_V) \right) = 1 - s_V.
$$

As $\lambda \to \infty$ we have $\alpha \to 0$ and thus $\mathcal{Z}_\lambda \to s_\mathcal{B}$. Since $s_\mathcal{B} \leq s_V$, we have $|s_V - \mathcal{Z}_\lambda| \to s_V - s_\mathcal{B}$ and $|\alpha s_V - \mathcal{Z}_\lambda| \to |0 - s_\mathcal{B}| = s_\mathcal{B}$, while the last term vanishes. Therefore,

$$
\lim_{\lambda \to \infty} D_{\mathrm{TV}}(\lambda, \varepsilon) = \frac{1}{2 s_\mathcal{B}} \left( q(s_V - s_\mathcal{B}) + (1 - q) s_\mathcal{B} \right) = \frac{1}{2 s_\mathcal{B}} \cdot 2 s_\mathcal{B}(1 - q) = 1 - q,
$$

where we used $s_\mathcal{B} = q \, s_V$ in the simplification.

**Existence of a calibrated $\lambda^\star$ and its closed form.** Observe that $\mathcal{Z}_\lambda$ is continuous in $\lambda$ and decreases from $1$ (at $\lambda = 0$) to $s_\mathcal{B}$ (as $\lambda \to \infty$). Since $s_\mathcal{B} \le s_V \le 1$, by the intermediate value theorem there exists $\lambda^\star \in [0, \infty]$ such that

$$\mathcal{Z}_{\lambda^\star} = s_V.$$

Equivalently, with $\alpha^\star := e^{-\lambda^\star}$,

$$s_V = s_\mathcal{B} + \alpha^\star(1 - s_\mathcal{B}) \quad \Longrightarrow \quad \alpha^\star = \frac{s_V - s_\mathcal{B}}{1 - s_\mathcal{B}} = \frac{s_V(1 - q)}{1 - q s_V}.$$

*Remark.* When $s_V > s_\mathcal{B}$ this gives a finite $\lambda^\star = \log \frac{1 - s_\mathcal{B}}{s_V - s_\mathcal{B}}$; if $s_V = s_\mathcal{B}$ then $\alpha^\star = 0$ corresponds to the limiting choice $\lambda^\star = +\infty$, which is consistent with the RLVR limit.

Under this calibration, the first absolute-value term vanishes: $|s_V - \mathcal{Z}_{\lambda^\star}| = 0$. To remove the remaining absolute value, note that for any $\alpha \in (0, 1]$,

$$\mathcal{Z}_\lambda - \alpha s_V = s_\mathcal{B} + \alpha(1 - s_\mathcal{B}) - \alpha s_V = \alpha(1 - s_V) + (1 - \alpha)s_\mathcal{B} \ge 0,$$

so $|\alpha s_V - \mathcal{Z}_\lambda| = \mathcal{Z}_\lambda - \alpha s_V$. Applying this at $\lambda^\star$ gives

$$|e^{-\lambda^\star} s_V - \mathcal{Z}_{\lambda^\star}| = |\alpha^\star s_V - s_V| = s_V(1 - \alpha^\star).$$

Substituting these identities into $D_{\text{TV}}$ (Equation (A4-12)) and using $\mathcal{Z}_{\lambda^\star} = s_V$ yields

$$D_{\text{TV}}(\lambda^\star, \varepsilon) = \frac{1}{2 s_V}\Big((1 - q) s_V(1 - \alpha^\star) + \alpha^\star(1 - s_V)\Big).$$

Finally, plug in $\alpha^\star = \frac{s_V(1-q)}{1 - q s_V}$ and $1 - \alpha^\star = \frac{1 - s_V}{1 - q s_V}$ to obtain the closed form

$$D_{\text{TV}}(\lambda^\star, \varepsilon) = \frac{(1 - q)(1 - s_V)}{1 - q s_V}.$$

**Strict improvement over the two limiting baselines.** From the closed form above,

$$\frac{D_{\text{TV}}(\lambda^\star, \varepsilon)}{1 - s_V} = \frac{1 - q}{1 - q s_V} \le 1 \quad \Longrightarrow \quad D_{\text{TV}}(\lambda^\star, \varepsilon) \le 1 - s_V,$$

and the inequality is strict whenever $q \in (0, 1)$ and $s_V \in (0, 1)$ (since then $1 - q s_V > 1 - q$). Similarly,

$$\frac{D_{\text{TV}}(\lambda^\star, \varepsilon)}{1 - q} = \frac{1 - s_V}{1 - q s_V} \le 1 \quad \Longrightarrow \quad D_{\text{TV}}(\lambda^\star, \varepsilon) \le 1 - q,$$

and it is strict for $q \in (0, 1)$ and $s_V \in (0, 1)$ (since then $1 - q s_V > 1 - s_V$). Therefore,

$$\boxed{D_{\text{TV}}(\lambda^\star, \varepsilon) \le \min\{1 - s_V, 1 - q\},}$$

with strict inequality in the non-degenerate interior regime.

**Monotone tightening *w.r.t.* $q$ under calibration.** Keeping $\lambda = \lambda^\star$ and treating $s_V$ as fixed, differentiate

$$D_{\text{TV}}(\lambda^\star, \varepsilon) = \frac{(1 - q)(1 - s_V)}{1 - q s_V}$$

with respect to $q$, we have:

$$\boxed{\frac{\partial}{\partial q} D_{\text{TV}}(\lambda^\star, \varepsilon) = (1 - s_V) \cdot \frac{-(1 - q s_V) + s_V(1 - q)}{(1 - q s_V)^2} = -\frac{(1 - s_V)^2}{(1 - q s_V)^2} < 0 \qquad (s_V \in (0, 1)).}$$

Hence the bound is strictly decreasing in $q$. Within the valid regime $\mathcal{B}_\varepsilon \subseteq \mathcal{S}_V$, enlarging $\varepsilon$ increases $s_\mathcal{B}$ and thus increases $q = s_\mathcal{B}/s_V$, which strictly tightens the bound. Under the regularity condition in Theorem 3.1, the inequality $q \ge \kappa(\varepsilon/\sigma)^{d_{\text{eff}}}$ further quantifies this monotone tightening as $\varepsilon$ increases while maintaining $\varepsilon \le \sigma$ so that $\mathcal{B}_\varepsilon \subseteq \mathcal{S}_V$ remains valid.

This completes the proof. □

# B. Implementation Details

## B.1. Dataset

**Overview.** To optimize PFlowNet, we curated a comprehensive training corpus by aggregating samples from large-scale open-domain multimodal VQA datasets, including the LLaVA (Liu et al., 2024a) official training set, VGR (Wang et al., 2025b), ArxivQA (Li et al., 2024b), VLM-R³ (Jiang et al., 2025), and ThinkLite-VL (Wang et al., 2025d). We first filtered the raw data based on task difficulty, typology, and evidence distribution, resulting in 95k visual-centric question-answer pairs. Specifically, a subset of 53k samples was processed via the pipeline described in Section 3.1 to generate perceptual flows; following multi-stage *verification* and *difficult control*, 45k high-quality samples were retained for cold-start initialization. The remaining 42k samples were reserved for the subsequent variational reinforcement fine-tuning stage. Notably, to ensure the effectiveness of evaluation, we rigorously cross-checked this corpus against the 15 adopted benchmarks to confirm *zero data overlap*, thereby minimizing the risk of data leakage.

*Table 5.* Training data construction via verifier-based filtering and difficulty-aware splitting. Here, $Z_\mathrm{s}$ denotes the synthetic perceptual flow, and $k_\mathrm{pass}$ denotes the *minimum* sampling budget required for the verifier to produce a correct answer, with $k_\mathrm{pass} > n$ indicating failure within $n$ decoded responses. In the data tuple, $E$ denotes the original expert RoIs before random expansion, and $Y$ denotes the *accepted response* generated by the verifier conditioned on $Z_\mathrm{s}$.

| Verification *w/o* $Z_\mathrm{s}$ | Verification *w* $Z_\mathrm{s}$ | Decision | Data Tuple |
|---|---|---|---|
| $k_\mathrm{pass} = 1$ | – | *Rejected* as trivial | – |
| – | $k_\mathrm{pass} > 1$ | *Rejected* as unverified flow | – |
| $2 \leq k_\mathrm{pass} \leq 16$ | $k_\mathrm{pass} = 1$ | *Accepted* to the RFT dataset | $(X, Y, E)$ |
| $k_\mathrm{pass} > 16$ | $k_\mathrm{pass} = 1$ | *Accepted* to the cold-start dataset | $(X, Z_\mathrm{s})$ |

**Verification & Difficulty Control.** After synthesizing candidate flows for all collected samples, we perform verifier-based filtering under two settings: (i) direct answering without the synthetic flow, *i.e.*, *w/o* $Z_\mathrm{s}$; and (ii) answering conditioned on $Z_\mathrm{s}$ and the corresponding zoomed-in evidence, *i.e.*, *w/* $Z_\mathrm{s}$. As summarized in Table 5, we first drop trivial samples and samples with unreliable flows, and then assign the remaining samples to either the cold-start set or the RFT set according to the performance gain induced by the synthetic grounding behaviors. Finally, the detailed statistics are provided in Figure 10.

**Cold Start.** For each sample $(X, Z_\mathrm{s})$ from the cold-start set, we initialize the policy via *supervised fine-tuning* by minimizing the cross-entropy loss between $p_\theta(Z \mid X)$ and the synthetic flow $Z_\mathrm{s}$. This teaches the policy to generate perceptual flows that benefit downstream reasoning.

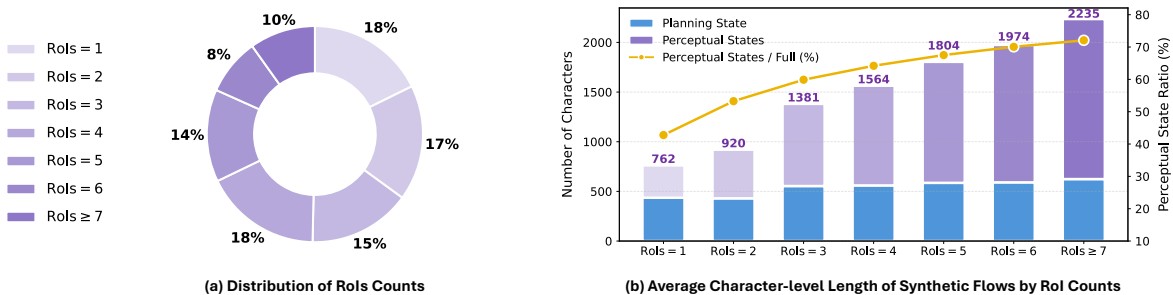

**(a) Distribution of RoIs Counts**    **(b) Average Character-level Length of Synthetic Flows by RoI Counts**

*Figure 10.* Statistics of the Cold-Start Dataset. Notably, as the number of RoIs increases, the average character length of the *Planning State* remains largely stable, whereas that of the *Perceptual States* grows substantially.

## B.2. Training Recipe

**Cold Start.** We initialize PFlowNet with Qwen3-VL-8B-Instruct (Bai et al., 2025a) and fine-tune it using the LLaMA-Factory framework (Zheng et al., 2024) on $16\times$ NVIDIA H200 GPUs. The model is trained on the 45k SFT samples for 3 epochs. We employ the AdamW optimizer (Loshchilov & Hutter, 2017) with a global batch size of 256 and a peak learning rate of $1 \times 10^{-5}$, employing a cosine decay schedule with a warm-up ratio of 0.1.

**RFT.** Initialized with the SFT checkpoint, PFlowNet is trained using a custom framework built upon the vLLM (Kwon et al., 2023) and TRL (von Werra et al., 2020) on $16\times$ NVIDIA H200 GPUs. We adopt a hybrid parallelism strategy to maximize throughput: data parallelism is applied across two nodes, DeepSpeed ZeRO-3 shards the policy parameters across GPUs

*Table 6.* Hyperparameters for variational reinforcement fine-tuning.

| Hyperparameter | Value | Hyperparameter | Value |
|---|---|---|---|
| Vicinal shaping intensity ($\lambda$) | 4.5 | Optimizer | AdamW |
| Vicinal radius ($\varepsilon$) | 0.5 | Peak learning rate | $5 \times 10^{-6}$ |
| Reward temperature | 1.0 | Weight decay | 0.05 |
| Exploration samples ($L$) | 8 | Warmup ratio | 0.02 |
| Sampling temperature (max) | 1.0 | Batch size per device | 2 |
| Sampling temperature (min) | 0.7 | Gradient accum. steps | 32 |
| Rollout Batch Size (sample-level) | 256 | Global Batch Size (response-level) | 1024 |
| Max flow length | 4,096 | Gradient clipping | 1.0 |
| Min flow length | 128 | Max input tokens | 16,384 |
| Image resolution (min pixels) | 3,670 | Image resolution (max pixels) | 12,845,056 |

within each node, and the reward model is fully replicated on each device to reduce communication overhead. Training is performed on 42k samples for 5 epochs, with detailed hyperparameters reported in Table 6.

### B.3. Exploration & Exploitation

We alternate between vLLM-based *rollout generation* and TRL-based *reward computation* and *policy optimization* in a serial manner. At each iteration, the current policy is first loaded into the vLLM engine to generate a rollout buffer, which is then consumed by the TRL-based trainer for reward computation and policy updates. Afterwards, the updated policy weights are synchronized back to the vLLM engine before generating the next rollout buffer. Notably, we employ the same system prompt, provided in Section B.5, for both training and self-conditioned reasoning. The special token $\langle$/localize$\rangle$ is used to apart the perceptual behaviors from the flow-conditioned reasoning. Specifically, during rollout, we treat $\langle$/localize$\rangle$ as a *custom stop token*: once this token is detected, the exploration process is terminated.

During the self-conditioned reasoning stage, we organize the input using the same system prompt, the original multimodal input, the generated perceptual flow, and the zoomed-in visual evidence targeted by the flow. The resulting conversation template is structured as follows:

| | |
|---:|---|
| system: | system prompt, |
| user: | multimodal input & zoomed-in visual evidence, |
| assistant: | generated perceptual flow. |

The model then continues generation conditioned on this structured context for final reasoning response.

### B.4. Reward Calculation

We employ *teacher forcing* to obtain outputs of the reward model $p_\phi$, and utilize the resulting logits to efficiently compute the RFT optimization objective defined in Equation (1). Specifically, treating each state as a token sequence, we calculate the transition probability $\log p_\theta(z_k \mid z_{0:k-1})$ by summing the autoregressive log-probabilities of the tokens within $z_k$. Given a data sample $(X, Y, E) \sim \mathcal{P}_{\text{data}}$ and a sampled relation $Z \sim p_\theta(Z \mid X)$, the computation involves three primary components:

(1) Transition probabilities: $\log p_\theta(z_k \mid z_{0:k-1})$ and $\log p_\theta(\top \mid z_{0:j})$;

(2) Efficacy reward: $\log p_\phi(Y \mid z_{0:k}, \top, X)$;

(3) Quality reward: the ratio $\log p_\phi^+(z_i) - \log p_\phi^-(z_i)$.

To eliminate redundant computations arising from shared prefixes in the first two components, we designed an efficient parallelization strategy. Specifically, we concatenate the shared flow with multiple terminal states or ground-truth labels. By leveraging customized position indices and attention masks (as illustrated in Figures 11 and 12), we compute all terms corresponding to the sub-flows within a single forward pass. Regarding the third component (quality reward), while it is intuitive to infer vision token indices from RoI coordinates to enable similar parallelization via dynamic masking, we identify two critical challenges. First, the resulting attention masks and position indices are often non-contiguous, leading to implementation complexity. Second, due to the native resolution property, the visual encoder in Qwen3-VL potentially resizes cropped inputs to enhance information density. Simply masking the original image tokens fails to replicate this process, thereby degrading the reward model's perceptual fidelity and compromising the accuracy of the reward calculation. Consequently, we explicitly crop the regions $I^+$ and $I^-$ and regard them as two separate inputs to the reward model for $\log p_\phi^+(z_i)$ and $\log p_\phi^-(z_i)$, thereby computing the ratio $\log p_\phi^+(z_i) - \log p_\phi^-(z_i)$.

## B.5. Prompt

**SYSTEM_PROMPT**

You are a helpful visual reasoning assistant. The user asks a question about an image, and you must provide a visually grounded answer by following a four-stage reasoning process in a fixed format. For every question, you must output the following four blocks in this exact order:

(1) Question analysis: analyze and interpret the user's question, clarify what needs to be recognized, counted, compared, or inferred from the image, and wrap this entire step in `<analyze></analyze>` tags;

(2) Evidence localization (interleaved): identify the image regions that are most helpful for answering the question, wrap the entire localization step in `<localize></localize>` tags, and inside `<localize>...</localize>` follow an interleaved pattern where for each region you first output the bounding box coordinates wrapped in `<box></box>` tags in the format `<box>[x1, y1, x2, y2]</box>` and then immediately explain how this region helps answer the question before moving on to the next region and repeating the same pattern;

(3) Evidence verification: review the previously localized regions, their corresponding explanations and supplied visual evidence (if available) to perform step-by-step reasoning, explicitly connect these visual evidence to the final conclusion, and wrap the entire reasoning process in `<thinking></thinking>` tags;

(4) Final answer: provide a clear, concise answer to the user's question without introducing new reasoning, and wrap the answer in `<answer></answer>` tags.

You must always include all four stages `<analyze>`, `<localize>`, `<thinking>`, and `<answer>`, keep the tag names and their order exactly as specified, ensure that the `<localize>` stage follows the interleaved pattern where each `<box>...</box>` is immediately followed by an explanation, and never output any text outside these four tagged blocks.

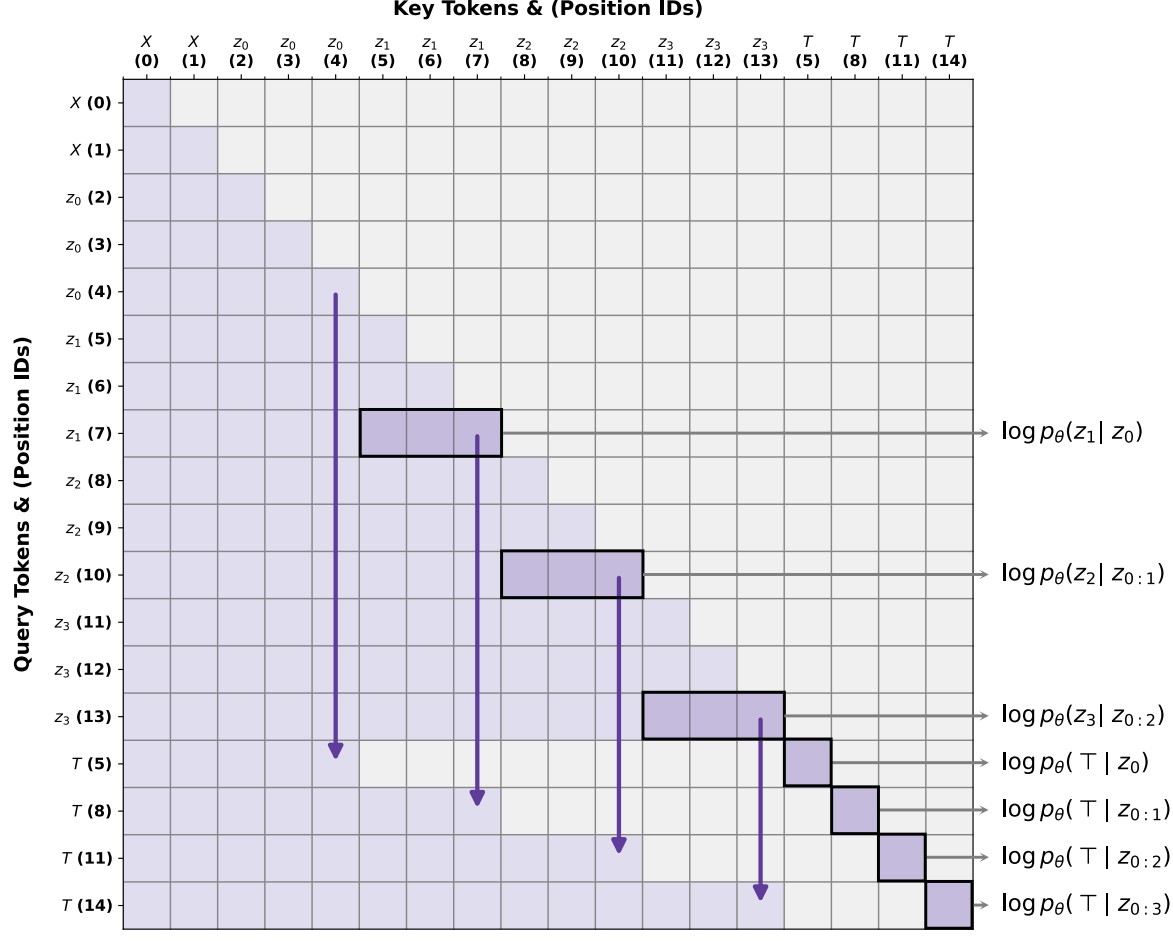

*Figure 11.* Parallel computation strategy of terminal probability, *i.e.*, $\log p_\theta(\top \mid z_{0:i})$, with explicit position IDs & attention mask.

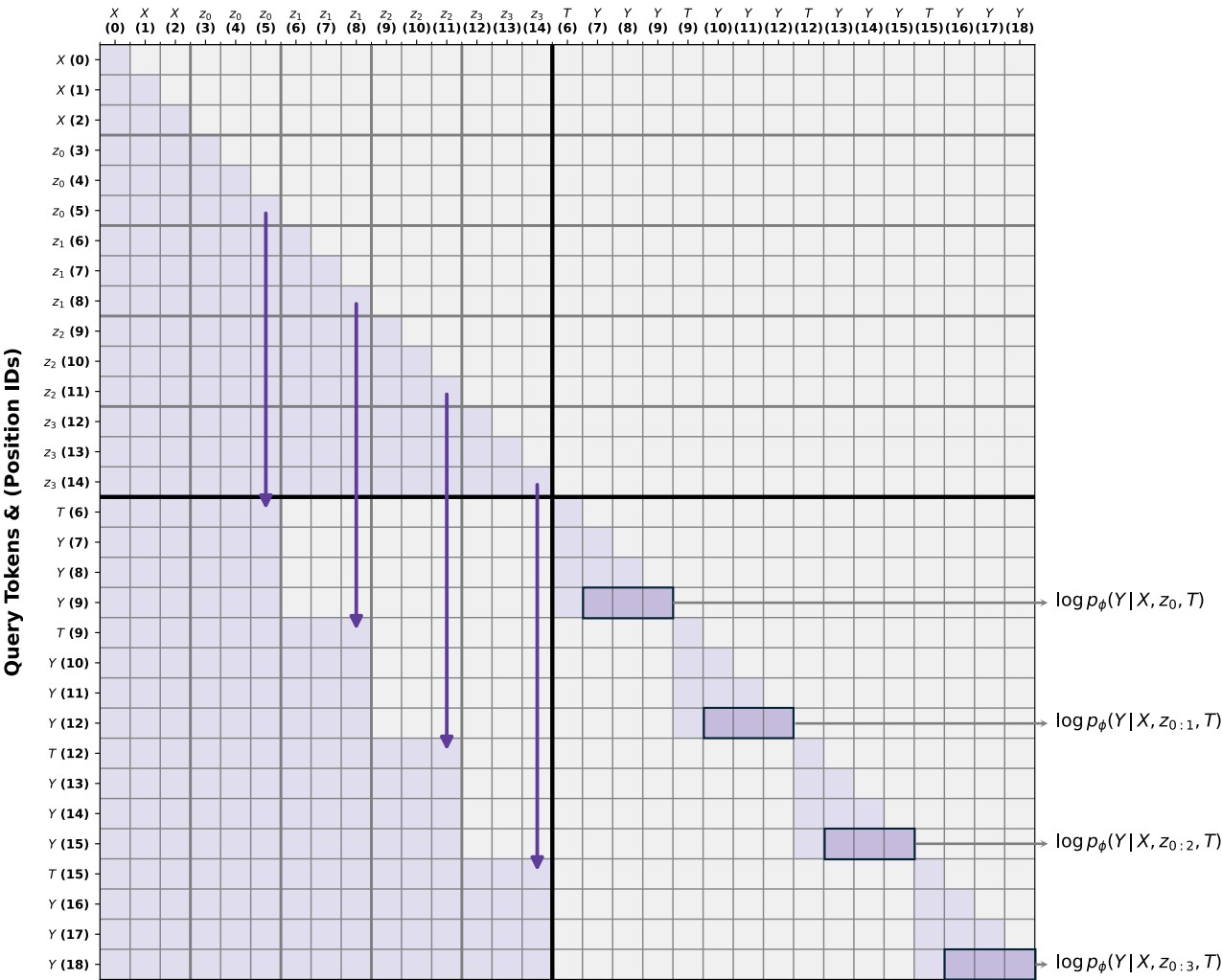

*Figure 12.* Parallel computation strategy of efficacy reward, *i.e.*, $\log p_\phi(Y \mid z_{0:i}, X)$, with explicit attention mask.

# C. Experimental Setup

## C.1. Benchmarks and metrics

Our evaluation targets visually grounded reasoning from two complementary angles: (i) *general-purpose* visual question answering (VQA) that measures broad perception, knowledge, and robustness without requiring explicit evidence localization; and (ii) *fine-grained* VQA/grounding benchmarks that stress high-resolution inputs, small targets, and explicit region-level evidence—precisely the regime where modeling *Perceptual Flow* is expected to be most beneficial. In total, we report results on 15 widely used benchmarks, following the default protocols from their original evaluations. Unless otherwise specified, we use accuracy for answer correctness; for grounded benchmarks we additionally report localization metrics (*e.g.*, mIoU) when annotated evidence is available.

**General-purpose VQA Benchmarks.** **MMBench$_{\text{dev}}^{\text{en}}$** (Liu et al., 2024d) provides a comprehensive multi-choice evaluation of multimodal capabilities, spanning fundamental perception, compositional understanding, and higher-level reasoning. Its structured design enables a fine-grained diagnosis of whether gains stem from improved perception or language-side inference. **MME-RealWorld-Lite** (Zhang et al., 2024) is a real-world variant of the MME-style evaluation, designed to reduce dataset bias and emphasize practical visual understanding. It covers diverse perception- and reasoning-centric skills (*e.g.*, OCR, document/scene perception, multi-object understanding), offering a robust stress test for real-world visual grounding. **POPE** (Li et al., 2023) focuses on *object hallucination* by asking binary questions about object presence. It directly quantifies the tendency of LVLMs to fabricate visual entities, making it a targeted benchmark for evaluating hallucination mitigation. **HallusionBench** (Guan et al., 2024) evaluates *detailed visual hallucination* via carefully constructed image-question pairs that probe object attributes, relations, and fine-grained semantics. Compared to coarse hallucination tests, it emphasizes subtle visual distinctions and consistency with the image. **AI2D$_{\text{test}}$** (Kembhavi et al., 2016) measures diagram understanding and elementary scientific reasoning over educational figures. It tests whether models can correctly interpret schematic structures, labels, and spatial relations rather than relying on natural-image priors. **ChartQA$_{\text{test}}$** (Masry et al., 2022) evaluates chart understanding, requiring models to extract numerical values, read legends/axes, and perform lightweight quantitative reasoning grounded in visual plots. **MathVision** (Wang et al., 2024) targets mathematical visual reasoning over figures (*e.g.*, geometry diagrams and math-centric illustrations). It assesses whether models can ground symbolic reasoning in precise visual cues, which is often brittle under language bias. **CV-Bench-2D / CV-Bench-3D** (Tong et al., 2024) is a vision-centric VQA suite repurposed from classic vision tasks to probe fundamental *2D* understanding (*e.g.*, spatial relations, counting) and *3D* understanding (*e.g.*, depth order) within a multimodal QA interface.

**Fine-grained VQA and Grounded Benchmarks.** **V\* Bench** (Wu & Xie, 2024) is a fine-grained visual search benchmark emphasizing small targets and localization-sensitive queries. It includes subsets such as *Attribute* and *Spatial* that require resolving subtle attributes or spatial configurations, where correct answers typically depend on identifying the right evidence region. **HR-Bench (4K/8K)** (Wang et al., 2025c) evaluates high-resolution VQA under long-context visual inputs. It contains both *Single* (single high-resolution image) and *Cross* (cross-image / cross-region) settings, stressing the ability to preserve fine details, track small objects, and aggregate evidence across large visual fields. **TreeBench** (Wang et al., 2025a) is a grounded reasoning benchmark that jointly evaluates *answer correctness* and *evidence localization quality* (mIoU). Its taxonomy separates *Perception* (*e.g.*, attributes, OCR, object retrieval) from *Reasoning* (*e.g.*, perspective transforms, ordering, comparisons), enabling a targeted analysis of whether a method improves perception behaviors, reasoning behaviors, or both. **ScreenSpot (v2 / Pro)** (Cheng et al., 2024; Wu et al., 2024; Li et al., 2025b) evaluates GUI grounding from screenshots: given an instruction, the model must localize the corresponding UI element (typically via point or box prediction). ScreenSpot-Pro further stresses professional software scenarios with high-resolution screens and smaller targets, making it a representative benchmark for visually grounded interaction and GUI understanding.

## C.2. Baselines

To validate the effectiveness of PFlowNet, we compare against (i) strong *general-purpose LVLMs* that provide competitive zero-/few-shot performance, and (ii) representative *visually grounded reasoning* approaches that explicitly model perception actions, which we categorize into *agentic frameworks* and *grounded RLVR* baselines (Section 5).

**General-purpose LVLM Backbones.** We include widely adopted instruction-tuned LVLMs (*e.g.*, InternVL3 (Zhu et al., 2025), Qwen2.5-VL (Bai et al., 2025b), Qwen3-VL (Bai et al., 2025a)) across multiple scales to control for backbone strength. We further report results from leading proprietary or frontier models (*e.g.*, GPT-4o/o3 (OpenAI, 2024; 2025a) and

Gemini variants (DeepMind, 2025a;b)) when available in the corresponding benchmark protocols, providing an upper-bound reference for general VQA and robustness.

**Agentic Frameworks.** Agentic frameworks enhance LVLMs with explicit interaction loops and external tools, typically coupling multi-turn planning with image operations (*e.g.*, Zoom-In), code execution, or sandboxed tool calls. **Thyme** (Zhang et al., 2025) represents "thinking with images" by allowing the model to write and execute code for visual processing, improving perception-heavy tasks at the cost of increased latency and tool dependency. **DeepEyes / DeepEyesV2** (Zheng et al., 2025b; Hong et al., 2025) are tool-augmented grounded reasoning systems that interleave language reasoning with explicit perceptual actions (*e.g.*, zoom/crop/inspect), often relying on external executors to stabilize evidence acquisition. **VACoT** (Xu et al., 2025) uses visual tools to mitigate performance degradation under challenging inputs (*e.g.*, low quality or ambiguous evidence), emphasizing tool-based intermediate visual steps. For GUI-centric evaluation, **Claude Computer Use** (Hu et al., 2024a) and **OpenAI CUA** (OpenAI, 2025b) serve as strong agentic baselines that integrate perception with action policies for computer-use settings, reflecting the state of practice for tool-using GUI agents.

**Grounded RLVR and Training-free Methods.** Grounded RLVR methods train policies with *verifiable* grounding-related rewards by representing perception as explicit spatial tokens (boxes/points) and optimizing the policy toward better evidence localization and answer correctness. **TreeVGR** (Wang et al., 2025a) is a representative grounded RLVR baseline on TreeBench-style tasks, coupling answer reward with localization supervision (often via IoU-style verifiers) to reduce language bias. **Pixel-Reasoner** (Su et al., 2025) performs multi-step region selection and refinement to acquire evidence for grounded reasoning, emphasizing iterative perception-to-reasoning transitions. **ZoomRefine** (Yu et al., 2025) adopts progressive zoom-in/refinement strategies to improve fine-grained evidence capture, typically benefiting attribute/OCR-like perception where small regions matter. **DyFo** (Li et al., 2025a) represents grounded optimization that encourages structured perceptual behaviors (*e.g.*, MCTS) to improve fine-grained understanding under constrained perception budgets.

**GUI Grounding Methods.** We additionally include explicit grounding models that predict clickable targets from screenshots: **SeeClick** (Cheng et al., 2024) is a screenshot-based GUI agent emphasizing GUI grounding pretraining and realistic element localization. **OS-Atlas** (Wu et al., 2024) is a foundation GUI action/grounding model trained on large-scale cross-platform GUI element corpora, outputting normalized coordinates for interaction targets. **UGround** (Gou et al., 2024) advocates a human-like, fully visual embodiment for GUI agents that perceive GUIs directly from pixels and act via pixel-level operations. **UI-TARS** (Qin et al., 2025) is an end-to-end native GUI agent model that operates directly on screenshots and produces human-like interaction outputs, serving as a strong modern baseline for GUI grounding.

## C.3. Evaluation Protocol

**Evaluation Framework.** To ensure a fair comparison, we reproduce all baseline results using their official evaluation pipelines with default configurations. Specifically, for the performance-efficiency and test-time scaling analyses, we migrated the Transformers-based implementations of *TreeVGR* and *Thyme* to *VLMEvalKit* (v0.1.0) utilizing the vLLM backend. For *DeepEyes*, we adopted its official pipeline, which is natively built on VLMEvalKit and vLLM. This standardization ensures strictly consistent experimental conditions, eliminating system-level discrepancies in latency and memory usage caused by different infrastructure frameworks. All evaluations were performed on an NVIDIA H200 GPU.

**Decoding Strategy.** For fairness, we employ greedy decoding for all models in standard evaluations. Conversely, for test-time scaling experiments, we utilize stochastic decoding to generate $k$ independent responses per sample. Specifically, pass@$k$ sampling is configured with `temperature=1.0` and nucleus sampling with `top-p=0.95` (no explicit `top-k` truncation is applied unless required by backend defaults).

**Prompting and Inference.** For all baseline methods, we adopt their official system prompts and templates (if available) to ensure optimal performance. PFlowNet utilizes the system prompt detailed in Section B.5. During inference, PFlowNet's generation is truncated immediately upon detecting the perceptual flow end-of-sequence token (`</localize>`). We then parse the Region of Interest (RoI) from the perceptual flow, extract the corresponding fine-grained visual features, and concatenate them with the initial perceptual flow to prompt the model for continued generation, thereby achieving self-conditioned autoregressive generation. Notably, we enforce the identical system prompt across both stages to ensure consistency between perceptual and reasoning behaviors.

# D. Additional Qualitative Analysis

## D.1. Analysis of Test-Time Scaling Behaviors

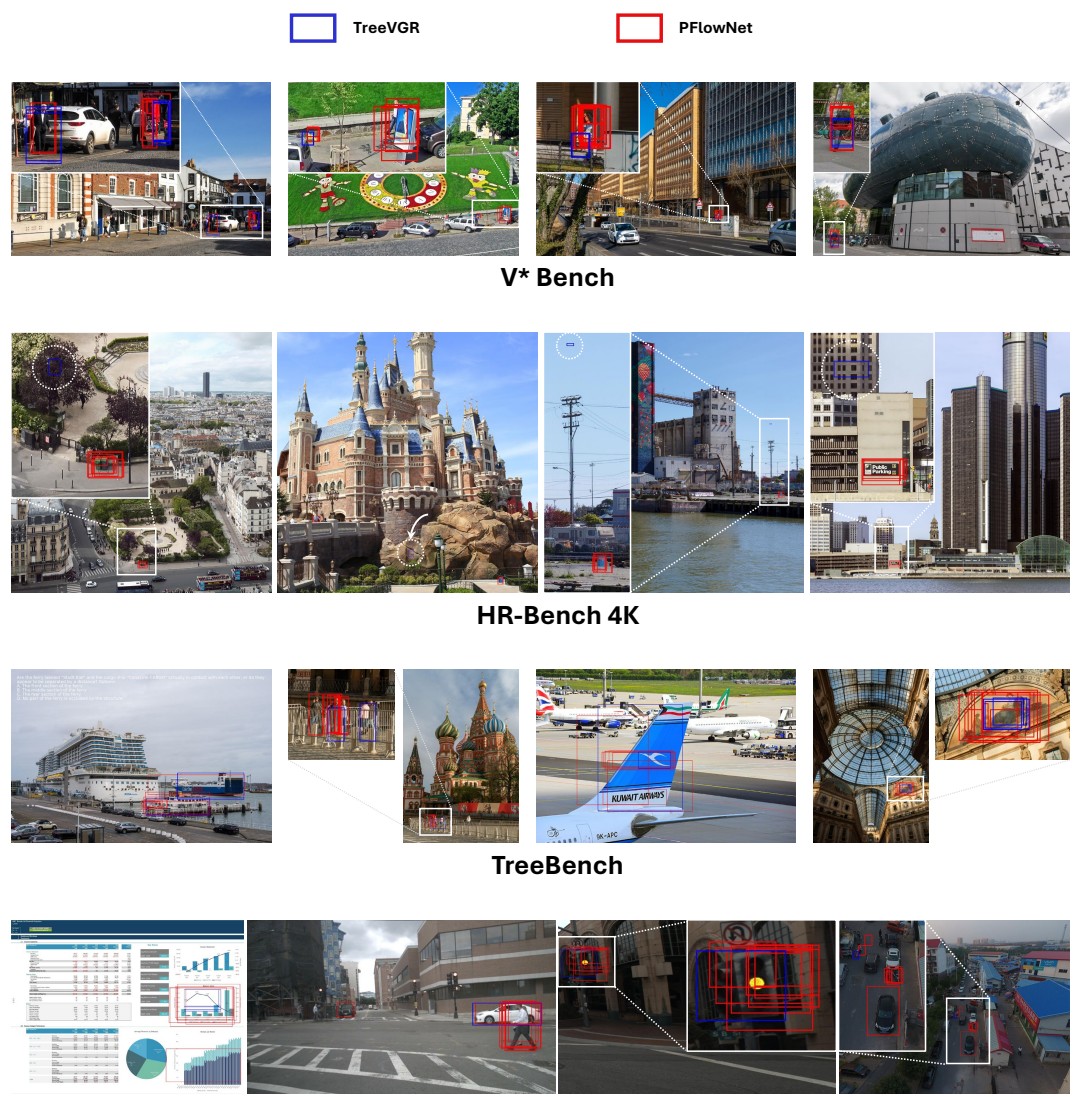

*Figure 13.* Qualitative comparison of grounding results under test-time scaling, highlighting the severe mode collapse in TreeVGR versus the diverse yet reliable perceptual exploration in PFlowNet. This visualization provides an intrinsic explanation for the results in Figure 5: as the computational budget increases, TreeVGR fails to sample diverse latent variables, thereby limiting effective likelihood gains.

To intuitively demonstrate the mode collapse near expert trajectories often exhibited by Grounded RLVR methods, we visualize grounding results selected from four benchmarks under the test-time scaling setting. As presented in Figure 13, for TreeVGR, *i.e.*, a representative Grounded RLVR method, the bounding boxes generated across multiple reasoning paths overlap almost entirely as the computational budget increases. This indicates a severe lack of perceptual diversity, preventing the model from attending to alternative visual regions even when such exploration is beneficial for reasoning.

In contrast, PFlowNet produces significantly more diverse Regions of Interest (RoIs) across multiple samples. This validates that approximating the target posterior via a variational objective is more effective than rigidly aligning with expert priors in mitigating collapse. Notably, TreeVGR exhibits severe hallucinations in the sample selected from the HR-Bench 4K and attends to featureless background regions. Crucially, due to its collapsed policy, the model lacks the capability to self-correct, persistently focusing on the same erroneous areas despite repeated computation.

## D.2. Analysis of Failure Case

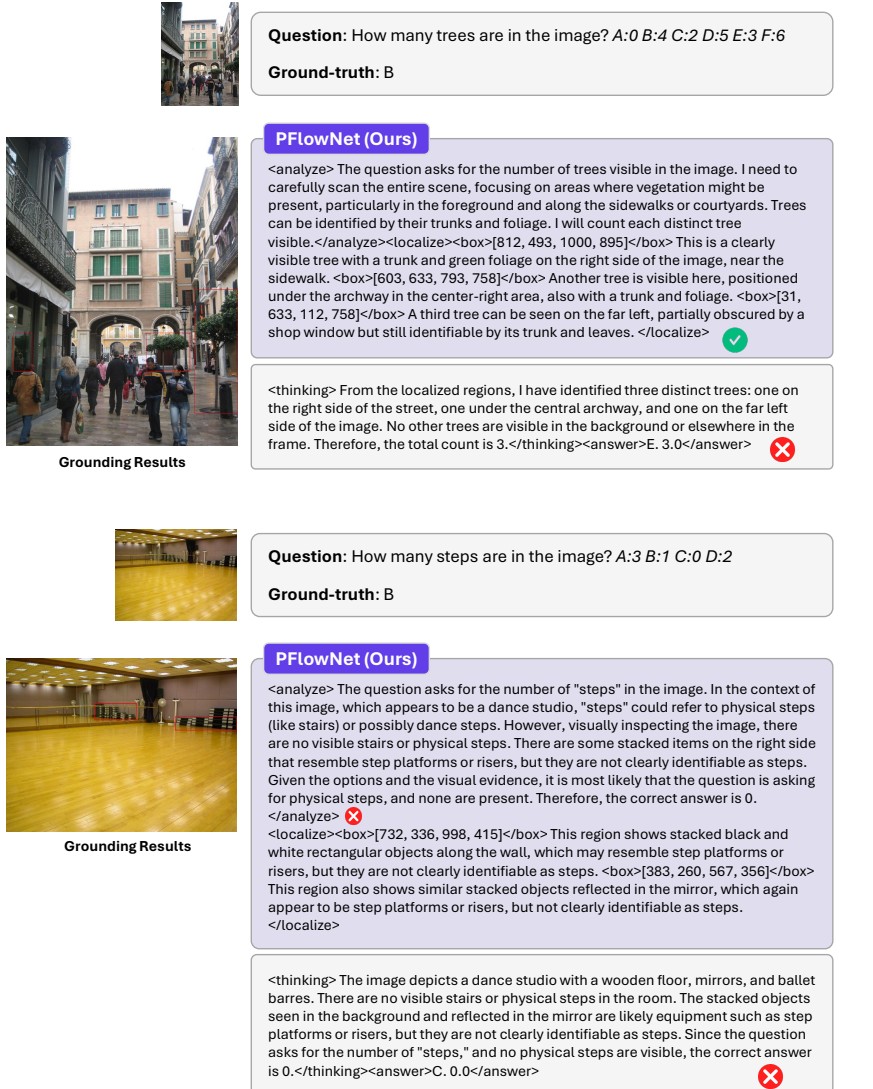

*Figure 14.* Additional qualitative results of visual reasoning. We highlight the important reasoning steps.

We also conduct an in-depth analysis of the failure cases in PFlowNet and identify two primary limitations.

First, a trade-off exists between geometric reliability and fine-grained counting. Since PFlowNet is incentivized to output diverse and reliable bounding boxes, it potentially merges spatially adjacent regions to preserve inter-object context. While it may be beneficial for general visual tasks, this behavior can lead to errors in counting tasks, where the model can be biased by the number of boxes in the perceptual flow. Crucially, as discussed in our ablation study (Table 3), the perceptual flow exerts a strong priming effect on the subsequent reasoning process; consequently, this issue cannot be fully mitigated by simply supplementing fine-grained visual features.

Second, the *planning state* lacks explicit supervision in our current framework, relying solely on passive optimization via the sub-flow level *Efficacy* term in the reward Equation (3). As a result, in challenging scenarios, *e.g.*, OOD scenarios, the model may fail to correctly decompose the necessary evidence. This decomposition failure propagates downstream, inevitably resulting in confusing perceptual behaviors and incorrect reasoning processes.

Addressing these challenges remains a primary focus for our future work.

## D.3. More Examples

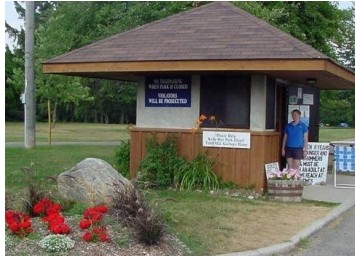

> **Question**: How many roads are in the image? A:2 B:3 C:1 D:0
>
> **Ground-truth**: A

**Qwen3-VL 8B**

<thinking>I need to count the number of roads visible in the image. Looking at the scene, I can see a paved surface that runs along the front of the building and extends into the distance on the right side. This appears to be one road. ***There is also a gravel or dirt path visible on the left side of the image***, which seems to be a separate path or driveway leading away from the main area. ***However, this is not clearly a road for vehicles. The most prominent and clearly defined road is the paved one***. Therefore, there is 1 road in the image.</thinking><answer>C</answer>

❌ *Language Bias*

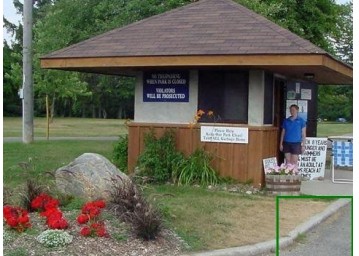

**Grounding Results**

**TreeVGR**

<think> First, identify the road-like elements in the image. <box>[375,275,480,375]</box> shows a paved path that could be considered a road. ***There are no other such elements visible***. So, there is only one road-like path in the image.</think> <answer>C</answer>

❌ *Missing Evidence*

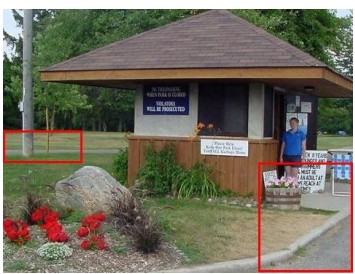

**Grounding Results**

**PFlowNet (Ours)**

<analyze> The question asks to count the number of roads visible in the image. A road is typically a paved or surfaced path for vehicles or pedestrians. I need to identify and count all distinct road surfaces or pathways shown in the image. </analyze><localize><box>[730, 584, 998, 997]</box> *This is a paved surface with a curb and a sidewalk, which is clearly a road or driveway leading to the building*. <box>[0, 461, 227, 584]</box> *This is another paved area in the **background**, likely a road or parking lot, visible behind the trees and utility pole*.</localize>

<thinking> *I have identified two distinct paved areas that can be classified as roads*: **one in the foreground** (the paved path with the curb) and **one in the background** (the paved area behind the trees). There are no other visible roads or paved paths in the image. Therefore, the total count is 2.</thinking><answer>A</answer> ✅

*Figure 15.* Additional qualitative results of visual reasoning. We highlight the important reasoning steps.

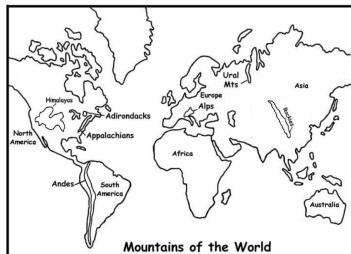

**Question**: *Based on the map,* are the Rocky Mountains located in Asia?

**Ground-truth**: Yes

**Qwen3-VL 8B**

<thinking>***The map shows the Rocky Mountains labeled in the continent of North America, specifically to the west of the United States***. Asia is shown to the east of North America on the map, and the Rocky Mountains are not located there. Therefore, the answer is no.</thinking><answer>***no***</answer> ❌ *Hallucination*

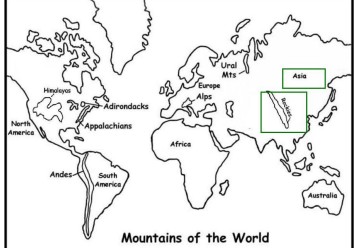

**Grounding Results**

**TreeVGR**

<think>First, identify the Rocky Mountains <box>[664,252,784,358]</box> on the map. They are in the western part of North America. Then, look at the label "Asia" <box>[720,192,832,244]</box> which is in the eastern part of the map. ***Since the Rocky Mountains are in North America and Asia is in the opposite side of the map, they are not in Asia.***</think> <answer>***No***</answer> ❌ *Language Bias*

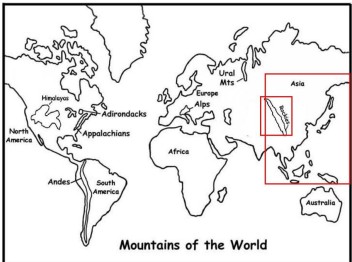

**Grounding Results**

**PFlowNet (Ours)**

<analyze> The question asks whether the Rocky Mountains are located in Asia based on the provided map. I need to locate the ***Rocky Mountains*** on the map and identify the continent they are situated on, ***then compare that to Asia***. </analyze>
<localize> <box>[738, 356, 829, 504]</box> *This region shows the label "Rockies" pointing to a mountain range in the western part of the landmass labeled "Asia".* <box>[751, 270, 995, 688]</box> *This region highlights the large landmass labeled "Asia", which includes the area where the Rockies are drawn*. </localize>

<thinking> *The map clearly labels the "Rockies" within the boundaries of the landmass labeled "Asia".* However, ***geographically***, the Rocky Mountains are located in North America, not Asia. The map appears to be inaccurate or misleading in its labeling. ***Based on the visual evidence presented in the map itself***, the Rockies are shown to be within Asia according to this diagram. Therefore, following the map's own labeling, the answer is yes. </thinking><answer>***yes***</answer> ✅

*Figure 16.* Additional qualitative results of visual reasoning. We highlight the important reasoning steps.

