# OpenReview forum: "Perceptual Flow Network for Visually Grounded Reasoning"
_ICML.cc/2026/Conference — ICML 2026 regular_

### Official Review · Reviewer_rEeu · 2026-03-11

**Soundness:** 3
**Presentation:** 3
**Significance:** 3
**Originality:** 3
**Overall Recommendation:** 4
**Confidence:** 2

**Summary:**

PFlowNet proposes a new framework for visually grounded reasoning in large vision-language models by arguing that rigid supervision from expert bounding boxes is often geometrically precise but not truly optimal for reasoning. It introduces a structured perceptual flow, which explicitly models the reasoning process as a planning state followed by a sequence of grounded perceptual states, each containing a region of interest and its caption. Based on this design, the method decouples perception from reasoning and trains the model with supervised fine-tuning plus a variational reinforcement fine-tuning strategy that combines multi-dimensional rewards with vicinal geometric shaping. This allows the model to explore perceptual behaviors that are both visually reliable and more useful for downstream reasoning, while also improving interpretability.

**Compliance With Llm Reviewing Policy:**

Affirmed.

**Final Justification:**

I think the reviewer’s rebuttal has addressed most of my concerns, so I am inclined to accept.

**Key Questions For Authors:**

- How much of PFlowNet’s performance depends on the availability of expensive external supervision during training, such as annotated RoIs, teacher-generated trajectories, and verifier-based rejection sampling, and how well would the method scale to domains where such resources are unavailable or much noisier?
- To what extent do the gains come from improved visual grounding itself, versus from using perceptual flow as a structured textual/semantic scaffold for autoregressive reasoning?

**Limitations:**

Yes

**Strengths And Weaknesses:**

Although I am not from this specific field, I can basically understand the contributions, and I believe the work is generally in line with the standard of an accepted ICML paper.

Strengths:
- The paper identifies a very concrete and well-motivated problem. Instead of simply assuming that expert bounding boxes are always beneficial, the authors first show that the most geometrically precise evidence is not necessarily the most useful for reasoning, and they explicitly connect this to a [tunnel vision effect] caused by overly tight localization.
- PFlowNet introduces a clear factorization between perception and reasoning, formalizes a perceptual flow with a planning state and perceptual states, and combines this with variational RFT, multi-dimensional rewards, and vicinal geometric shaping. These components are tied together by a consistent objective instead of feeling like isolated heuristics.
- The paper reports gains on fine-grained benchmarks such as V* Bench, HR-Bench, and ScreenSpot, and on general-purpose reasoning benchmarks such as TreeBench and MME-RealWorld-Lite. It also includes efficiency and test-time scaling analyses, which strengthens the claim that the method is practically useful rather than only accurate on a narrow setup.

Weakness:
- The training pipeline still depends heavily on external supervision and filtering. The paper explicitly states that determining the golden flow remains inherently challenging, which suggests the method does not fully escape expensive external supervision.
- The authors report that adding external fine-grained visual features brings only marginal improvements, whereas removing the perceptual flow causes severe degradation; they further interpret the flow as an explicit semantic anchor. This is interesting, but it also leaves some ambiguity about whether the main contribution is improved visual grounding or a particularly effective structured textual conditioning mechanism.

---

> ### Author Rebuttal · Authors · 2026-03-25
>
> Thanks for the thoughtful review. To address your concerns, we provide a point-by-point response below.
>
> ***
>
> >**W1**. The training pipeline still depends heavily on external supervision and filtering, e.g., annotated RoIs, teacher-generated trajectories, and verifier-based rejection sampling.
>
> **A**. We agree that the SFT stage requires teacher-based trajectory synthesis and filtering. However, we would like to emphasize two points: **1)** The RFT stage, another core component of our method, only uses coarse RoI boxes for vicinal geometric shaping. **2)** PFlowNet still yields clear performance gains over the base model even without expensive teacher-generated trajectories (**see response to Q1 2)**).
>
> > **Q1 1)**. How much of PFlowNet's performance depends on the availability of expensive external supervision during training?
>
> **A**. Actually, our ablation (Sec. 4.3, Table 2, rows 1/2/5) separates the gains from SFT and RFT:
>
> ||TreeVGR|V* Bench|MME-RealWorld-Lite|
> |---|:---:|:---:|:---:|
> |Base Model|44.9|77.5|46.0|
> |+SFT|48.3|83.7|54.2|
> |$\Delta$ vs. Base|+3.4|+6.2|+8.2|
> |+RFT|55.3|90.6|67.0|
> |$\Delta$ vs. SFT|**+7.0**|**+6.9**|**+12.8**|
>
> These results show that SFT based on dense expert trajectories is helpful, but a large fraction of the final gain comes from the variational RFT stage, which *does **not** require trajectory-level supervision.*
>
> > **Q1 2)**. How well would the method scale to domains where such resources are much noisier or even unavailable?
>
> **A**. We note that PFlowNet has two design choices improving robustness. First, rejection sampling filters noisy captions and hallucinations in trajectories. Second, annotated RoIs are used only as anchors defining a reward vicinity, not as fixed gold targets, making PFlowNet tolerant to moderate localization errors.
>
> Considering your concerns, *we further try to directly optimize the base model, i.e., Qwen3-VL 8B, via the proposed variational RFT paradigm **without SFT***, and compare it with the full pipeline. We keep the main training recipe the same as in Appx. B, and extend RFT from 5 epochs to 15 epochs as its early rollouts are less stable without SFT.
> ||SFT|TreeVGR|V* Bench|MME-RealWorld-Lite|
> |---|:---:|:---:|:---:|:---:|
> |Qwen3-VL 8B|-|44.9|77.5|46.0|
> |+ RFT (5 ep)|✔︎|55.3|90.6|67.0|
> |$\Delta$ vs. Base||+10.4|+13.1|+21|
> |+ RFT (5 ep)|✘|47.5|81.4|52.8|
> |+ RFT (15 ep)|✘|53.8|88.5|62.5|
> |$\Delta$ vs. Base||+8.9|+11|+16.5|
>
> We observe that, even without dense supervision, PFlowNet can yield **large gains** over the base model (+8.9 / +11.0 / +16.5) by scaling up optimization compute. We attribute this to the strong grounding prior already present in Qwen3-VL, which is pretrained on large-scale grounding data and post-trained with think-with-images capabilities [1]. Thus, even without expert trajectory distillation, structured perceptual behaviors can still **emerge** under our carefully designed prompt (Appx. B.4) and RFT paradigm. *These results demonstrate the potential of our method when the expensive teacher-generated trajectories are unavailable.*
>
> [1] Bai S, et al. Qwen3-vl technical report. arXiv preprint arXiv:2511.21631.
>
> ---
>
> >**W2**. Ambiguity about whether the main contribution is improved by visual grounding or structured textual conditioning.
>
> >**Q2**. To what extent do the gains come from improved visual grounding itself, versus from using perceptual flow as a structured textual/semantic scaffold for autoregressive reasoning?
>
> **A**. From an empirical perspective, our ablation (Sec. 4.3, Table 2, rows 5/7/8) directly probes this. Here, "*Textual*" denotes the perceptual flow and "*Visual*" denotes the zoomed-in evidence used during the reasoning stage:
>
> | |Textual|Visual|TreeVGR|V* Bench|MME-RealWorld-Lite|
> |-|:-:|:-:|:-:|:-:|:-:|
> |PFlowNet|✘|✔︎|49.2|83.8|52.1|
> |PFlowNet|✔︎|✘|54.5|89.4|66.4|
> |PFlowNet|✔︎|✔︎|55.3|90.6|67.0|
>
> These results suggest that the gain does **not** mainly come from simply appending localized visual crops as extra visual input. Specifically, if we keep only the grounded visual evidence but remove the perceptual flow, performance drops substantially to 49.2 / 83.8 / 52.1. In contrast, keeping the textual perceptual flow while removing the zoomed-in visual evidence retains **approximately 99%** of the full model's performance. Therefore, the dominant gain appears to come from **grounded semantic anchoring**.
>
> *We believe this is related to the **later-fusion** design of most LVLMs.* Intuitively, since these models are built by **extending pretrained LLMs** with vision encoders and cross-modal alignment, *it is **easier for them to exploit structured textual conditions** than additional raw visual evidence.* From this perspective, PFlowNet can be understood as learning to *transcribe relevant multimodal information into an explicit semantic anchor,* thereby better facilitating downstream reasoning.
>
> ---
>
> We hope the above clarifications are helpful to address your concerns.

---

> > ### Author Rebuttal · Reviewer_rEeu · 2026-04-04
> >
> > Thank the authors for the rebuttal and the extra analysis. I think it addresses my concerns. And I would like to maintain my score.

---

> > > ### Author Response · Authors · 2026-04-04
> > >
> > > Dear reviewer rEeu,
> > >
> > > We sincerely appreciate that our feedback addresses your concerns. Your insightful comments are very important to further strengthen this work.
> > >
> > > Best regards,
> > >
> > > Submission 7377 Authors.

---

### Official Review · Reviewer_kBhx · 2026-03-11

**Soundness:** 3
**Presentation:** 4
**Significance:** 3
**Originality:** 3
**Overall Recommendation:** 5
**Confidence:** 2

**Summary:**

This paper introduces Perceptual Flow Network (PFlowNet), a framework designed to improve the visually grounded reasoning of Large Vision-Language Models (LVLMs).

The core premise is that "perfect" geometric boxes (like those from object detectors) are often suboptimal for reasoning because they lack necessary context (the "tunnel vision" effect). PFlowNet instead allows the model to learn "reasoning-oriented" perceptual behaviors—finding the visual evidence that actually helps answer the question, even if it deviates slightly from strict expert annotations.

**Compliance With Llm Reviewing Policy:**

Affirmed.

**Final Justification:**

The rebuttal solve my questions.

**Key Questions For Authors:**

N/A

**Strengths And Weaknesses:**

**Strengths:**

- Mitigates Hallucination & Bias: By anchoring the reasoning process in specific visual evidence ("Self-conditioned reasoning"), it reduces the likelihood of the model "guessing" based on language patterns.

- Superior Performance: It sets new State-of-the-Art (SOTA) records on challenging benchmarks, specifically V* Bench (90.6%) and MME-RealWorld-lite (67.0%), significantly outperforming the base Qwen3-VL 8B model.

- Interpretability: The "flow" provides a clear, step-by-step trace of what the model looked at and why, which is easier for humans to verify than a standard "black box" CoT.

- Test-Time Scaling: Unlike previous methods that plateau quickly, PFlowNet shows "robust test-time scaling," meaning the model’s accuracy continues to improve if you allow it more "thinking" or sampling time during inference.

**Weaknesses:**

- Latency: While more efficient than multi-turn agentic tools, the decoupled "perceive-then-reason" flow still adds computational overhead compared to a standard single-pass LVLM.

- Dependence on "Expert" Priors: Although it avoids rigid alignment, the model still relies on having some initial expert annotations (geometric priors) to define the "vicinity" for exploration. It may struggle in domains where no such detection experts exist.

---

> ### Author Rebuttal · Authors · 2026-03-25
>
> Thank you for the helpful comments. We provide a point-by-point response below.
>
> ---
> >**W1. Latency:** While more efficient than multi-turn agentic tools, the decoupled “perceive-then-reason” flow still introduces computational overhead compared with a standard single-pass LVLM.
>
> **A1.** We agree that the performance–latency trade-off is an important and highly practical factor for LVLMs; therefore, we have already analyzed this issue in Sec. 4.2 (Fig. 6 and Table 9). To further address this concern, we clarify two additional points:
>
> 1) PFlowNet **does not modify the underlying LVLM architecture**, e.g., by introducing additional layers or external modules. The extra latency mainly comes from the longer context, i.e., the generated perceptual flow and the visual tokens corresponding to fine-grained evidence.
>
> 2) Unlike most existing methods, PFlowNet is optimized in a **decoupled** manner. It *learns to produce a multimodal prefix that facilitates high-quality downstream reasoning, rather than directly altering the LVLM's native reasoning pathway*. Therefore, PFlowNet can **largely preserve the model's reasoning behavior** even without the perceptual flow.
>
> As a result, *PFlowNet supports flexible inference via prompting*: **Perceive-then-Reason** (P-then-R) with the prompt introduced in Appx B.4; and **Direct Reasoning** (Direct R) that *skips the first perception stage* via the standard reasoning prompts used in Qwen3-VL. To validate these claims from an empirical perspective, we further report both accuracy and latency under these two modes below.
> ||Qwen3-VL 8B|PFlowNet(P-then-R)||PFlowNet(Direct R)||
> |---|---:|---:|---:|---:|---:|
> |**Benchmark**|Acc (%) / Latency (s)|Acc (%) / Latency (s)|Δ Acc / Latency|Acc (%) / Latency (s)|Δ Acc / Latency|
> |V* Bench|79.1 / 0.40|90.2 / 1.70|+11.1 / +1.30|80.9 / 0.45|+1.8 / +0.05|
> |HR-Bench 4k|72.4 / 0.70|80.5 / 2.20|+8.0 / +1.50|72.1 / 0.74|-0.3 / +0.04|
> |HR-Bench 8k|68.1 / 3.22|76.8 / 4.90|+8.8 / +1.68|67.5 / 3.35|-0.6 / +0.13|
> |CV-Bench-2D|79.5 / 0.10|87.9 / 1.30|+8.4 / +1.20|78.2 / 0.14|-1.3 / +0.04|
> |MME-RealWorld-Lite|48.6 / 1.10|67.6 / 2.60|+19.0 / +1.50|50.5 / 1.12|+1.9 / +0.02|
> |MMBench-dev-en|85.3 / 0.04|91.4 / 1.52|+6.1 / +1.48|84.9 / 0.08|-0.4 / +0.04|
> |AI2D-test|84.0 / 0.05|90.1 / 1.65|+6.1 / +1.60|85.2 / 0.10|+1.2 / +0.05|
> |MathVision|25.2 / 0.90|32.3 / 2.70|+7.1 / +1.80|27.4 / 1.03|+2.2 / +0.13|
>
> As shown in the table, P-then-R **consistently brings substantial accuracy gains** (typically ~ 6% – 10%, up to 19%) at the cost of **moderate additional latency** (roughly 1.2–1.8s on most benchmarks), whereas direct reasoning remains largely comparable to the base Qwen3-VL 8B with negligible additional latency. This shows that *PFlowNet does not impose a fixed inference penalty*; instead, it offers a tunable accuracy–efficiency trade-off.
>
> ---
>
> >**W2**. **Dependence on "Expert" Priors**: Although it avoids rigid alignment, the model still relies on initial expert annotations (geometric priors) to define the "vicinity" for exploration. It may struggle in domains where such detection experts do not exist.
>
> **A**. Thank you for raising this important point. We agree that domains without usable detection priors are more challenging for visually-grounded reasoning methods, since such methods typically require some auxiliary signal to *constrain the intermediate visual thoughts of LVLMs during reasoning*. This is **not unique to PFlowNet**; many prior works also rely on similar priors in different forms, such as **Thyme** (NeurIPS'25), which *requires image-manipulation code with RoIs*; **ViGoRL** (NeurIPS'25), which *requires the central points of RoIs*; **DeepEyes** family (ICLR'26), which *requires long-horizon trajectories with RoIs*; **PixelReasoner** (NeurIPS'25) and **TreeVGR** (ICLR'26), which *require bounding boxes of RoIs*.
>
> However, PFlowNet places a weaker requirement on expert priors than these approaches. Unlike agentic frameworks that depend on multi-turn fine-grained trajectories involving tool use or code execution, PFlowNet **only requires coarse RoI bounding boxes** as geometric priors, which **can often be obtained from general detectors**. More importantly, these priors are not used as fixed supervision targets. Instead, they are used only to define a local vicinity for perceptual exploration. As a result, *PFlowNet places a lower accuracy requirement on expert priors, and is less sensitive to moderate prior errors within the local exploration region (detailed in Appendix E.1), rather than rigidly inheriting the expert’s mistakes.*
>
> Therefore, while we agree that domains with no usable detection prior remain a limitation, we believe PFlowNet has broad practical applicability in realistic settings where coarse RoIs are available, and its dependence on expert priors is milder than that of prior frameworks for visual grounded reasoning.
>
> ---
>
> We hope the above clarifications address your concerns.

---

> > ### Author Rebuttal · Reviewer_kBhx · 2026-04-02
> >
> > Thanks the authors for their rebuttal. I'd like to raise the score (4 -> 5).

---

> > > ### Author Response · Authors · 2026-04-02
> > >
> > > Dear reviewer kBhx,
> > >
> > > We are sincerely grateful for your positive feedback and for the time dedicated to re-evaluating our manuscript. Your insightful suggestions have significantly improved the quality and clarity of this work.
> > >
> > > Best regards,
> > >
> > > Submission 7377 Authors.

---

### Official Review · Reviewer_KhXm · 2026-03-11

**Soundness:** 3
**Presentation:** 3
**Significance:** 3
**Originality:** 3
**Overall Recommendation:** 4
**Confidence:** 3

**Summary:**

The paper proposed a perceptual flow network that combines guided reasoning chain in the visual reasoning process. The author found that having guided reasoning chain grounds the visual reasoning process and showed it via several experiments and comparisons with baseline methods.

**Compliance With Llm Reviewing Policy:**

Affirmed.

**Final Justification:**

My biggest concern before is that this method is similar to CoVT and would add very little value to the research community. But after going through author's rebuttal response with additional experiment analyzing the advantages of PFlowNet over CoVT, as well as their comparison between PFlowNet and CoVT, my concerns were resolved and I can now see the value of this work. Therefore I would like to raise my score from 2->4.

**Key Questions For Authors:**

Q1: what is the difference between this method (perceptual flow) and visual chain-of-thought method like [1]? My understanding is that this method provides a ground-truth annotated ROI flow (perceptual flow) during the training to ground the visual reasoning. Isn’t this the ROI in the visual chain-of-thought that models pay attention to? If so I’m having a hard time understanding how is the proposed perceptual flow grounds visual reasoning better than visual CoT.

Q2: Is it possible to have confidence interval for the reported results?

**Limitations:**

Yes

**Strengths And Weaknesses:**

Strength:

S1: The paper is working on an important problem, and the math is solid.

Weakness:

W1: The difference between this method and previous published methods are not clear. See Q1.

W2: The reported improvements are limited and having delta with vanilla at the last line of the table 1 is misleading. Also no confidence interval is reported in any reported data, which could be really helpful to identify any significant improvement, given that the reported improvements are minimal by absolute value.

W3 (Minor): Although the math is solid, it is a little bit unnecessarily complicated. Consider combining some parts maybe? (Like definition 2.2 with section 3)

Reference:

[1] Qin, Yiming, et al. "Chain-of-visual-thought: Teaching vlms to see and think better with continuous visual tokens." arXiv preprint arXiv:2511.19418 (2025).

---

> ### Author Rebuttal · Authors · 2026-03-25
>
> Thanks for your comments. To address your concerns, we provide a point-by-point response:
>
> ---
>
> >**W1 & Q1**. The difference between PFlowNet and the prior method: *"Chain-of-Visual-Thought (CoVT), Qin, Y, et al."*, is unclear.
>
> **A**. We agree that CoVT is a relevant work, as both methods aim to improve VLM reasoning; however, *CoVT and PFlowNet differ substantially in five aspects*.
>
> **1. Visual-thought representation**. CoVT uses four types of **learnable special tokens** (segmentation, depth, edge, and DINO tokens) to encode visual thought. In contrast, PFlowNet externalizes visual thought as RoIs with captions via **structured text tokens** (Def. 2.2), *yielding a directly interpretable reasoning process without additional decoders.*
>
> **2. Role of RoIs**. CoVT directly treats DINO-generated RoIs as **fixed targets**, *which can inherit the biased geometric priors of the detector (Prelim Study, Sec. 1)*. By contrast, PFlowNet *avoids direct supervision on detector-provided RoIs* and instead uses them only as anchors to define a **reward vicinity**, *which enables performance-driven yet reliable visual reasoning.*
>
> **3. Training paradigm**. CoVT fully relies on **supervised fine-tuning** with multiple external experts (e.g., SAM, DepthAnything, PIDInet, and DINO). PFlowNet instead derives its gains mainly from **reinforcement fine-tuning** with a carefully designed multi-dimensional reward, rather than directly depending on expert priors (Table 2).
>
> **4. Utilization of evidence**. CoVT adopts self-generated special tokens to *compress the features of the visual evidence*, i.e., mask, edge, etc., while PFlowNet *directly acquires zoom-in evidence views* from the intermediate perceptual flow through an agentic two-stage pipeline. *This particularly benefits PFlowNet for fine-grained or high-resolution visual understanding* (e.g., V* / HR Bench). Conversely, *CoVT has a natural advantage on spatial reasoning tasks such as CV-Bench Depth / Dist because it explicitly models depth.*
>
> **5. Performance profile**. To enable a direct comparison, we also instantiate PFlowNet on *Qwen2.5-VL-7B-Instruct*, following the training recipe and dataset described in Appx B. The comparison results are shown as follows:
> ||V*|HR 4k|HR 8k|CV-Bench *Depth*|CV-Bench *Dist*
> :-|:-:|:-:|:-:|:-:|:-:
> Qwen2.5-VL 7B|76.4|68.6|64.9|72.8|75.5
> CoVT|78.5|72.5|69.9|**89.2**|**82.5**
> PFlowNet|**88.5**|**78.1**|**74.1**|81.4|80.3
>
> Overall, CoVT and the proposed PFlowNet exhibit **complementary advantages** in different types of tasks, validating their difference in design principles from an empirical perspective. Considering your comment, we have added this discussion in the revised manuscript (Appx C.2) to highlight the originality and scope of PFlowNet.
> >**W2 1)**. The improvements over baselines are limited.
>
> **A**. To directly answer the concerns about the performance of PFlowNet, we reorganize the main results from Tables 1, 5, 6, and 9 as follows:
> ||Venues|V*|HR 4k|HR 8k|TreeBench|MME-rw-Lite|MMB$_{EN}$|HallusionBench|AI2D|MathVision
> :-|:-|:-:|:-:|:-:|:-:|:-:|:-:|:-:|:-:|:-:
> Thyme|NeurIPS'25|82.2|77.0|72.0|38.2|54.4|81.1|55.6|79.7|26.7
> TreeVGR|ICLR'26|85.8|76.5|72.0|50.4|54.9|79.2|66.4|80.5|24.7
> DeepEyes V2|ICLR'26|81.8|77.9|73.8|40.7|52.4|76.3|42.1|80.3|26.0
> PFlowNet|-|90.6|80.5|76.8|55.3|67.0|91.4|73.6|90.1|32.3
> $\Delta$ vs. *the second best*|-|**+4.8**|+2.6|+3.0|**+4.9**|**+12.1**|**+10.3**|**+7.2**|**+9.6**|**+5.6**
>
> As shown above, PFlowNet delivers **consistent gains** over strong recent baselines across a broad set of representative benchmarks, with improvements typically in the **3%–10%** range and exceeding 10% on several tasks. In our view, these gains are *broad, non-trivial, and practically meaningful.*
> >**W2 2)**. "$\Delta$ vs. vanilla" in Table 1 is misleading.
>
> **A**. We clarify that "$\Delta$ vs. vanilla" was *originally used to denote the gain of PFlowNet over the base Qwen3-VL-8B.* To avoid potential ambiguity, in the revised manuscript we have directly merged Qwen3-VL-8B into the General LVLMs rows and replaced the last row with "$\Delta$ vs. the second best".
> >**W2 3) & Q2**. No confidence interval is reported for main results.
>
> **A**. We clarify that the benchmarks are fixed, the model outputs are **decoded greedily**, and their correctness is **directly verifiable**. Thus, the results are **near-deterministic**. We nevertheless *run each benchmark three times and report the mean to account for minor system-level noise.*  (Evaluation Protocol, Appx C.3). Following your suggestion, *we have also added confidence intervals for the results in the revised manuscript.*
> >**W3**. The math is solid while somewhat unnecessarily complicated.
>
> **A**. Following your suggestion, we have streamlined the notation in the sec. 3 and moved its secondary details to the appendix, preserving the technical soundness while making the core insight more transparent.
>
> ---
>
> Sincerely, we hope our responses are helpful to address your concerns.

---

> > ### Author Rebuttal · Reviewer_KhXm · 2026-04-03
> >
> > I would like to thank the reviewers for addressing the weaknesses and questions I listed in the original review! I especially appreciate the author's response to my question regarding "CoVT vs PFlowNet", and I think the comparison really shows the advantages of CoVT and PFlowNet respectively.
> >
> > I see the value of this work could bring to the community now and therefore would like to raise my score (2->4). Thank you very much for the response!

---

> > > ### Author Response · Authors · 2026-04-03
> > >
> > > Dear reviewer KhXm,
> > >
> > > We are sincerely grateful for your positive feedback and for the time dedicated to re-evaluating our manuscript! Your insightful suggestions have significantly improved the quality and clarity of this work.
> > >
> > > Best regards,
> > >
> > > Submission 7377 Authors.

---

### Official Review · Reviewer_iPVm · 2026-03-12

**Soundness:** 3
**Presentation:** 3
**Significance:** 2
**Originality:** 2
**Overall Recommendation:** 4
**Confidence:** 3

**Summary:**

This paper proposes PFlowNet, a training framework for visually grounded reasoning in LVLMs. The method introduces a structured latent trajectory called a perceptual flow, which explicitly decouples visual perception from reasoning. The model first samples a visual evidence trajectory from the perceptual flow distribution and then generates the final answer conditioned on that trajectory. PFlowNet is optimized using a variational reinforcement learning objective based on sub-trajectory balance, allowing reward signals to propagate through intermediate states of the trajectory rather than relying solely on sparse terminal rewards. The authors further design a multi-dimensional reward function, including a contrastive evidence-quality term and an answer-likelihood term that measures reasoning effectiveness. In addition, the paper introduces a vicinal geometric shaping mechanism, which constrains exploration within a neighborhood of expert geometric priors, allowing controlled exploration while maintaining visual reliability. PFlowNet is evaluated on several visual reasoning and visual understanding benchmarks. Experimental results show that PFlowNet achieves strong performance on reasoning-oriented visual tasks.

**Compliance With Llm Reviewing Policy:**

Affirmed.

**Final Justification:**

The response addresses my concerns

**Key Questions For Authors:**

1. I would like to know whether the distribution of flow lengths during inference differs from the standard flow length distribution used during training. Does the model tend to generate flows that are significantly longer or shorter than those seen in training?

2. How sensitive is the method to noise or missing expert priors E? If the expert detection boxes are inaccurate, can the model still train stably?

3. Since the reward uses the ratio p+/p−, is it possible that the model attempts to minimize p− rather than maximize p+ in order to increase the reward? For example, the model might try to make the complement of the RoI as small as possible so that it contains little object information.

**Limitations:**

yes

**Strengths And Weaknesses:**

Strengths

- One key strength of this work is the introduction of perceptual flow as a structured latent representation. By explicitly modeling the visual evidence acquisition process as a trajectory consisting of planning states and perceptual states, the model can iteratively select relevant visual regions and generate evidence descriptions during reasoning.

- The use of sub-trajectory balance allows reward signals to propagate through intermediate states of the trajectory instead of relying solely on final-answer supervision. Furthermore, the proposed multi-dimensional reward design jointly considers evidence quality and answer correctness. The contrastive term p+/p− discourages the model from relying purely on contextual guessing and instead encourages the use of visual evidence from the selected regions.

- The vicinal geometric shaping mechanism defines a Chamfer-IoU neighborhood around expert detection boxes and softly penalizes trajectories that deviate excessively from these priors. This may provide a principled way to balance exploration and reliability.

- The paper evaluates the approach on multiple visual reasoning and visual understanding benchmarks and includes various analyses.

Weaknesses

- Although the authors state that the method targets visually grounded reasoning, the RoI contrastive strategy may only apply to certain types of tasks. It seems that, for tasks such as mathematical reasoning, perceptual flows may not be directly useful. Therefore, the proposed approach may not generalize equally well to broader reasoning tasks.

- There seems to lack a description of data generation pipeline for the synthetic perceptual flows used during training, which makes it difficult to assess reproducibility and fairness of the experimental results.

- The length of perceptual flows in the dataset is unclear. Providing statistics on the distribution of flow lengths would help readers better understand how the flow mechanism operates in practice.

---

> ### Author Rebuttal · Authors · 2026-03-25
>
> Sincerely appreciate your thoughtful comments. We respond point by point below.
>
> ---
>
> >**W1**. Generalization beyond visually intensive tasks, e.g., mathematical reasoning.
>
> **A**. Appx. D.6 (Table 9) shows that PFlowNet also improves multimodal mathematical reasoning: on MathVision, it gains **+7.1** over the base Qwen3-VL-8B and outperforms strong baselines:
> ||Venues|MathVision
> :-|:-|:-:
> Qwen3 VL 8B||25.2
> Thyme|NeurIPS'25|26.7
> TreeVGR|ICLR'26|24.7
> DeepEyes V2|ICLR'26|26.0
> PFlowNet||**32.3**
> $\Delta$ vs. Base||+7.1
>
> We attribute this to the factor that PFlowNet first acquires relevant evidence and then conditions reasoning on it. This turns the multimodal evidence into a structured semantic anchor, which helps suppress irrelevant context and mitigate hallucination.
> >**W2**. Lack a description of data generation pipeline for the synthetic perceptual flows, making it difficult to assess reproducibility and fairness of the results.
>
> **A**. We clarify that the data-generation pipeline is described in the manuscript: **Sec. 3.1** introduces *Flow Synthesis* (L.199-214), **Fig. 4** shows the full synthesis pipeline, and **Appx. B.1** details the *data sources*, including a cross-check to ensure *zero overlap* with evaluation benchmarks. Following your suggestion, we have added the exact teacher prompts and more visualized examples in the revised manuscript. As stated in **Appx. F.2**, we will also release the codebase and curated SFT/RFT datasets to further strengthen reproducibility.
> >**W3**. The length of perceptual flows in the dataset is unclear.
>
> >**Q1**. I would like to know whether the distribution of flow lengths during inference differs from the standard flow length distribution used during training. Does the model tend to generate flows that are significantly longer or shorter than those seen in training?
>
> **A**. We computed the distribution of RoI counts and the corresponding average flow lengths (character level) for both the *synthesized trajectories*:
> RoIs|Percent(%)|Full|Analysis|Localize
> :-:|:-:|-:|-:|-:
> 1|17.8|762|436|326
> 2|17.4|920|430|490
> 3|15.4|1,381|554|827
> 4|17.6|1,564|560|1,004
> 5|13.9|1,804|586|1,218
> 6|8.5|1,974|591|1,383
> |>=7|9.9|2,235|624|1,611
>
> and the *model-generated flows* on two representative benchmarks:
> RoIs|Percent(%)|Full|Analysis|Localize
> :-:|:-:|-:|-:|-:
> *V* Bench
> 1|58.4|403|258|144
> 2|38.7|528|256|272
> |>=3|2.8|652|282|370
> *MME-RealWorld-Lite*
> 1|63.9|485|311|174
> 2|17.6|585|338|247
> 3|5.8|639|349|290
> 4|3.6|782|387|394
> |>=5|3.4|1,063|435|628
>
> For the same RoI count, generated flow lengths are highly similar across benchmarks, suggesting that flow length is mainly governed by the amount of required evidence rather than benchmark difficulty. Besides, compared with synthetic expert flows, model-generated flows are **generally shorter**. We attribute this to the contrastive term $(p^+/p^-)$ that makes the *grounded captions become shorter and more discriminative*.
> >**Q2**. How sensitive is the method to noise or missing expert priors E? If the expert detection boxes are inaccurate, can the model still train stably?
>
> **A**. We agree that training would become more challenging if expert priors were entirely missing or severely corrupted. However, PFlowNet uses trajectory-level **rejection sampling** to filter incorrect priors (Sec. 3.1), and treats expert priors only as anchors that define a reward vicinity, **not as fixed gold targets**, making it more tolerant to moderate localization errors.
> >**Q3**. Is it possible that the model attempts to minimize p− rather than maximize p+ in order to increase the reward (p+/p-)? For example, the model try to make the complement of the RoI as small as possible so that it contains little object information.
>
> **A**. We clarify that such *reward hacking* is **explicitly discouraged** by *vicinal geometric shaping* (Sec. 3.2, L.213–241). The final reward is$$R_\lambda(z_{0:k}\top):=\Big[\prod_{i=1}^k\frac{p_\phi^+(z_i)}{p_\phi^-(z_i)}\ p_\phi(Y\mid z_{0:k}\top,X)\Big]\textcolor{purple}{\omega_\lambda(z_{0:k},E)},$$
> where
>
> $$\omega_\lambda(z_{0:k},E):=e^{-\lambda\ \mathbb{I}(z_{0:k}\notin\mathcal{B}_\varepsilon(E))},$$
>
> $\mathcal{B}_\varepsilon(E):=\lbrace z\_{0:k}\mid d\_{\rm IoU}(r\_{1:k},E)\le\varepsilon \rbrace.$
>
> If the model enlarges the RoI so its complement becomes too small, then $d_{\rm IoU}(r_{1:k},E)>\varepsilon$, giving $\omega_\lambda(z_{0:k},E)=e^{-\lambda}$ and exponentially reducing the reward. Thus, the objective is not favored by suppressing $p_\phi^-(z_i)$ via trivially enlarging RoIs; it explicitly penalizes such lazy grounding. Furthermore, we also conduct an ablation of this:
> p+|p-|TreeVGR|V* Bench|MME-RealWorld-Lite
> :-:|:-:|:-:|:-:|:-:
> ✘|✔︎|50.5|84.8|58.2
> ✔︎|✘|52.2|88.1|65.5
> ✔︎|✔︎|55.3|90.6|67.0
>
> Empirically, maximizing $p_\phi^+(z_i)$ is the **primary driver** of the gains, while suppressing $p_\phi^-(z_i)$ further sharpens the caption to be more specific to the RoIs.
>
> ---
>
> We hope these clarifications address your concerns.

---

> > ### Author Rebuttal · Reviewer_iPVm · 2026-04-03
> >
> > Thank the authors for the rebuttal and the extra analysis. I think it addresses my concerns. And I would like to maintain my score.

---

> > > ### Author Response · Authors · 2026-04-03
> > >
> > > Dear reviewer iPVm,
> > >
> > > We sincerely appreciate that our feedback addresses your concerns. Your insightful comments are very important to further strengthen this work.
> > >
> > > Best regards,
> > >
> > > Submission 7377 Authors.

---

### Decision · Program_Chairs · 2026-04-30

**Decision:**

Accept (regular)

**Comment:**

All reviewers found the proposed PFlowNet and the concept of “perceptual flow” to be a well-motivated and effective solution for visually grounded reasoning. The reviewers specifically noted the method’s ability to mitigate hallucinations by anchoring reasoning in structured visual evidence, as well as its strong performance on fine-grained benchmarks. The rebuttal successfully addressed initial concerns regarding the novelty relative to prior work like CoVT and the reproducibility of the synthetic data generation pipeline, leading to a consensus for acceptance.

The authors are encouraged to strengthen the final version by incorporating reviewer suggestions, particularly by providing a more detailed description of the data generation pipeline for synthetic perceptual flows and including statistical distributions of flow lengths. The final manuscript should also integrate the additional experimental analyses provided during the rebuttal, specifically the comparison with CoVT, to clearly delineate the technical contributions. Finally, the authors should address the “tunnel vision effect” and the role of vicinal geometric shaping more prominently to clarify how the model balances exploration with reliability in complex visual domains.